# Application of Heterogeneous Catalytic Ozonation for Refractory Organics in Wastewater

**Bing Wang [1,2,*], Huan Zhang [1], Feifei Wang [3], Xingaoyuan Xiong [1], Kun Tian [1], Yubo Sun [1] and Tingting Yu [4]**

[1] School of Chemistry and Chemical Engineering, Southwest Petroleum University, Chengdu 610500, China; zhanghuan@stu.swpu.edu.cn (H.Z.); xiongxgy@stu.swpu.edu.cn (X.X.); tiankun@stu.swpu.edu.cn (K.T.); 201822000226@stu.swpu.edu.cn (Y.S.)

[2] State Key Laboratory of Oil and Gas Reservoir Geology and Exploitation, Southwest Petroleum University, Chengdu 610500, China

[3] College of Chemistry and Chemical Engineering, Xi'an Shiyou University, Xi'an 710065, China; wangfeifei10614@163.com

[4] Safety & Environment & Technology Supervision Research Institute of Southwest Oil & Gas Field Company, PetroChina, Chengdu 6100041, China; yutt@petrochina.com.cn

* Correspondence: wangb@swpu.edu.cn; Tel.: +028-8303-7303

**Abstract:** Catalytic ozonation is believed to belong to advanced oxidation processes (AOPs). Over the past decades, heterogeneous catalytic ozonation has received remarkable attention as an effective process for the degradation of refractory organics in wastewater, which can overcome some disadvantages of ozonation alone. Metal oxides, metals, and metal oxides supported on oxides, minerals modified with metals, and carbon materials are widely used as catalysts in heterogeneous catalytic ozonation processes due to their excellent catalytic ability. An understanding of the application can provide theoretical support for selecting suitable catalysts aimed at different kinds of wastewater to obtain higher pollutant removal efficiency. Therefore, the main objective of this review article is to provide a summary of the accomplishments concerning catalytic ozonation to point to the major directions for choosing the catalysts in catalytic ozonation in the future.

**Keywords:** AOPs; heterogeneous catalytic ozonation; mechanisms; ozonation

---

## 1. Introduction

As we know, water is commonly used in many industries as a solvent or reaction medium. In recent years, a significant number of chemical industries were likely to discharge untreated or partially-treated industrial wastewater in order to get a maximum profit, leading to an increase of wastewater volume. The organics in bulk solution, including refractory, toxic, carcinogenic and mutagenic organic compounds, have been increasing, which are harmful to humans, the environment, animals, etc. It is difficulty for those organic compounds to find ways to convert into innoxious substances, because most organics are chemically stable and non-biodegradable. Water demand is increasing day-by-day, thus solving percent conversion of organic compounds has become a severe problem and a priority [1].

At present, numerous chemical manufacturing industries use conventional wastewater treatment techniques, such as adsorption, coagulation, flocuulation, incineration, biological oxidation, etc. to treat wastewater. However, it was reported that conventional treatment methods have some limitations, which may lead to an inefficient removal efficiency in the treatment of recalcitrant organics. These conventional treatment methods are also costly due to the large space requirement and long

degradation time [2]. Facing those limitations, many scientific fraternities try their best to explore more advanced technologies to completely degrade refractory organic compounds.

Advanced oxidation processes (AOPs) have received increasing attention and are applied to degrade various organic compounds. AOPs refer to the generation of hydroxyl radicals (HO·), which are powerful, non-selective chemical oxidants and have a high efficiency in removing organic compounds from various kinds of wastewater at a laboratory scale. Among all AOPs, Fenton or Fenton-like, hydrogen peroxide [3], ozone based, photocatalysis [3], electrochemical oxidation (EAOP), catalytic wet peroxide oxidation (CWPO), and hydrodynamic and acoustic cavitation combined with AOPs [4] are extensively used. Ozonation processes promote only the partial oxidation of organic compounds in wastewater and sometimes it produces toxic intermediates [5]. Therefore, ozonation is put into practice in the presence of a catalyst, which is called catalytic ozonation, and has emerged as an effective treatment technology to achieve a higher degradation efficiency for all types of organic matters in the effluent [6].

In catalytic ozonation processes, many metal oxides, metals or metal oxides on supports, activated carbon and minerals are used as catalysts to improve the removal efficiency of the organic contaminants. This review paper emphasizes the application of catalysts in the catalytic ozonation processes to point to the major directions for choosing the catalyst in the catalytic ozonation for the future.

## 2. Advanced Oxidation Processes (AOPs)

Advanced oxidation processes (AOPs) are defined as oxidation of organic compounds occur primarily through reactions with hydroxyl radicals (HO·) [7]. AOPs have many of advantages than these other advanced treatment processes are short of degrading organic compounds in water. Table 1 presents a summary of advantages and disadvantages of AOPs [8].

**Table 1.** Advantages and disadvantages of AOPs for treatment of organic contaminants in wastewater.

| Advantages |
| --- |
| • Rapid reaction rates for most organic pollutants |
| • Degradation of pollutant, rather than concentrating it |
| • No production of solid residuals |
| • No need for regeneration to maintain process efficiency |
| • Ability to completely mineralize most contaminants |
| • Small footprint |
| • Not selective, capable of degrading virtually any contaminant |
| **Disadvantages** |
| • Can produce unknown transformation products |
| • Background water quality can interfere with removal efficiency |
| • May need to provide subsequent process to treat residual oxidant |

Hydroxyl radicals have a higher oxidizing capacity than other oxidizing agents (Table 2) [9], which could react rapidly with most organic compounds from wastewater. However, among those AOPs technologies, many technologies have some limitations, such as photo-catalytic oxidation has a low reaction rate, and it is costly due to the irradiation of catalysts being uniform, which results in a significant loss of energy. The catalysts in wet air oxidation reactions [10] are likely to deactivate due to poisoning, coke deposition, metal leaching [11], and electrocoagulation requires minimum solution conductivity [12]. Hence, ozone based AOPs are increasing popular, attractive, and promising. There are some advantages and disadvantages of different reinforcement ozone oxidation technologies and applications. The $H_2O_2/O_3$ process has a high efficiency in removing pollutants and has no by-products, but it consumes a high energy and there exists $H_2O_2$ residual [13,14]. The low utilization rate of energy and a lesser treatment volume exist in ultrasonic/$O_3$ system. In cavitation-based processes, there are some advantages, such as a high disinfecting rate and disadvantages, such as cavitation erosion [15–17]. Catalytic ozonation processes have emerged as an efficient treatment method to make wastewater biodegradable and less toxic, but the catalyst life and recyclability are problems.

**Table 2.** A comparison of the oxidizing potential of various oxidants.

| Oxidizing Agent | Electrochemical Oxidation Potential (EOP) (V) | EOP Relative to Chlorine |
|---|---|---|
| Fluorine | 3.06 | 2.25 |
| Hydroxyl radical | 2.80 | 2.05 |
| Oxygen atomic | 2.42 | 1.78 |
| $TiO_2$+hv [1] | 2.35 | 1.72 |
| Ozone | 2.08 | 1.52 |
| Persulfate | 2.01 | 1.48 |
| Perbromate | 1.85 | 1.35 |
| Hydrogen peroxide | 1.78 | 1.30 |
| Perhydroxyl radical | 1.70 | 1.25 |
| Hypochlorite | 1.49 | 1.10 |
| Bromate | 1.48 | 1.09 |
| Chlorine | 1.36 | 1.00 |
| Dichromate | 1.33 | 0.98 |
| Chlorine dioxide | 1.27 | 0.93 |
| Permanganate | 1.24 | 0.91 |
| Oxygen (molecular) | 1.23 | 0.90 |
| Perchlorate | 1.20 | 0.89 |
| Bromine | 1.09 | 0.80 |
| Iodine [2] | 0.54 | 0.39 |

[1] $TiO_2$+hv creates positive hole via electron excitation; [2] Iodine owns a strong ability to take electrons.

## 3. Heterogeneous Catalytic Ozonation

Ozone is an effective oxidant ($E° = +2.07$ eV), which can selectively oxidize unsaturated double bonds [18] (Figure 1) and aromatic structures [19]. Moreover, ozone may react with water and produce hydroxyl radicals which have a higher oxidation potential ($E° = +2.8$ eV). In general, ozone attacks organic compounds via two rates: A direct molecular ozone reaction and an indirect pathway.

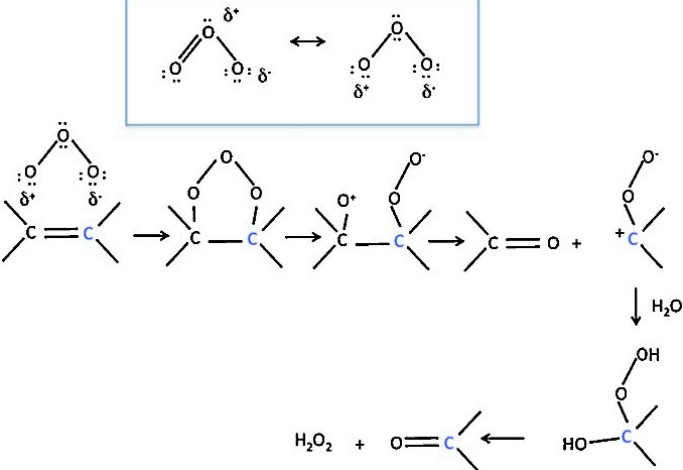

**Figure 1.** Schematic representation of direct $O_3$ reaction with unsaturated bond.

However, ozonation alone is inadequate for completely degrading organic pollutants due to its selectivity when it attacks some organic compounds [5]. The low solubility and instability of ozone in water leads to the low utilization rate.

As a consequence, ozonation is carried out with the addition of homogeneous or heterogeneous catalysts, called catalytic ozonation processes (COPs). COPs have emerged as an efficient treatment method to enhance the removal efficiency for all types of organic pollutants from the effluent [6]. The ozonation in the presence of a catalyst promotes the decomposition of ozone to generate hydroxyl radicals [20]. Various catalysts are utilized in organic compound treatments in COPs, such as alumina [21], activated carbon [22], natural minerals [23], and metal oxides [24–26]. These catalysts

can accelerate the decomposition of ozone to form free radicals (such as OH·), which further stimulates oxidation due to the high reaction rate [27,28]. Compared with homogeneous catalytic ozonation [29], heterogeneous catalytic ozonation is advantageous due to the sustained reactions via an active surface that does not require a continuous supply of reagents, and it is generally easy to retrieve the solid catalyst after a treatment. Therefore, seeking an active and stable catalyst is a vital task for environmental purposes and intensive studies have been made in recent years.

A variety of solid catalysts has been studied in the literatures. Among these catalysts, those that are widely used in heterogeneous catalytic ozonation are as follows:

- Metal oxides
- Metal or metal oxides on supports
- Carbon materials
- Minerals modified with metals

## 4. Metal Oxides

Various metal oxides play an important role in heterogeneous catalytic ozonation. Among them, $Al_2O_3$, $TiO_2$, $MnO_2$, FeOOH, and bimetallic oxides are studied as possible catalysts in heterogeneous catalytic ozonation. However, different studies suggested different ozonation mechanisms. The catalytic efficiency depends to a great extent on the catalyst and its surface properties and the pH of the solution that influences the properties of the surface-active sites (such as Lewis acid sites and Bronsted acid sites). Metal oxide surfaces have many hydroxyl groups, which effect the amount and the properties of hydroxyl. Figure 2 shows the variation of the surface hydroxyl species concentration with the surface pH [30].

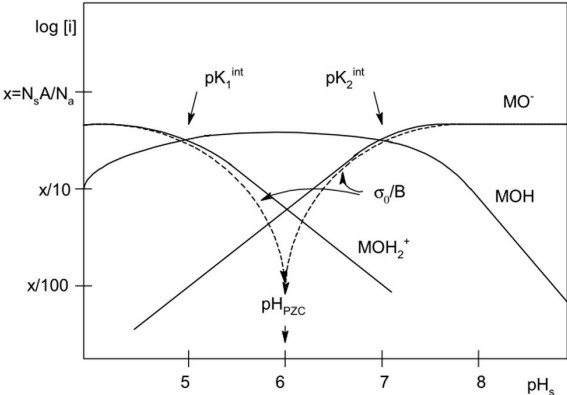

**Figure 2.** Variation of the surface hydroxyl species concentration with with the surface pH.

### 4.1. Manganese Oxide

Manganese oxide-based catalysts are considered to be the most effective metal oxide with a strong ability to react with gas phase ozone and they are promising active catalysts for heterogeneous catalytic ozonation in aqueous solutions [31–37]. Catalytic ozonation studies carried out using manganese oxide as catalyst are summarized in Table 3. Manganese oxide-based catalysts are believed to be effective and economical for ozone decomposition mainly due to their peculiar properties including high redox potential, low water solubility, environmental friendliness, ease of manufacture, low cost, diverse crystallographic structures, and different oxidation states [38,39]. Previous researchers reported [40] that commercial $MnO_2$ was not active in catalytic ozonation, but Luo et al. [41] reported that commercial $MnO_2$ had a catalytic effect on the degradation of Bisphenol A, but it played a certain inhibiting effect in the degradation of ibuprofen. The same result had been observed by Nawaz et al. [42] for the catalytic ozonation of 4-nitrophenol.

Nawaz et al. [43] used three different catalysts ($MnO_2$, $Mn_2O_3$, and $Mn_3O_4$) for the degradation of phenol (Ph), p-cresol (Ph-$CH_3$), and p-chlorophenol (Ph-Cl). At 60 min, all phenols could be completely decomposed over three catalysts. The degradation rates of all the phenols followed the order of $Mn_3O_4$ < $Mn_2O_3$ < $MnO_2$. $MnO_2$ was the most active among the three catalysts because of its higher surface hydroxyl groups and higher electron transfer ability. However, in the catalytic ozonation of benzene and toluene, the activity followed the order $MnO_2$ < $Mn_2O_3$ < $Mn_3O_4$, which was considered to be related to the surface area and oxygen mobility of manganese oxides. In the heterogeneous activation of peroxymonosulfate for the remove of phenol, the order was $MnO_2$ < $Mn_3O_4$ < $Mn_2O_3$, and this was considered to be connected with the redox potential of manganese oxides [44]. In the different kinds of wastewater, the same catalyst had different catalytic performances. They also found that superoxide radical ($^1O_2$), singlet oxygen ($^{\cdot}O_2{}^-$), and molecular ozone were mainly responsible for phenol degradation in bulk solution.

Tong et al. [40] investigated the heterogeneous catalytic ozonation of sulfosalicylic acid (SSal) and propionic acid (PPA) in aqueous solution, they found that the dependence of catalytic activity of $MnO_2$ on the pH of solutions and organic compounds, but it was independent of the type of $MnO_2$. For SSal, no catalytic activities of both $MnO_2$ (β-$MnO_2$ and γ-$MnO_2$) were observed at pH 6.8 and 8.5. When the value of pH was higher than zero point of charge (pH$_{zpc}$), $MnO_2$ almost had no catalytic efficiency during oxalic acid ozonation, which had been confirmed by Andreozzi et al. [45]. For PPA, no catalytic efficiency of different kinds of $MnO_2$ was observed at pH 1.0 and 6.8. Furthermore, they found that there was no direct relationship between the efficiency of metal oxide catalytic decomposition of ozone and that of its catalytic ozonation for the degradation of organic compounds.

Nawaz et al. [46] investigated the catalytic ozonation of 4-nitrophenol (4-NP) using six phases of $MnO_2$ (α-, β-, δ-, γ-, λ- and ε-) as catalyst. The removal efficiency of 4-NP and total organic carbon(TOC) increased in the order of α-$MnO_2$ (99.3%) > δ-$MnO_2$ (96.6%) > γ-$MnO_2$ (93.3%) > λ-$MnO_2$ (90.0%) > ε-$MnO_2$ (89.8%) > β-$MnO_2$ (86.4%), α-$MnO_2$ (82.4%) > δ-$MnO_2$ (73.5%) > γ-$MnO_2$ (64.2%) > λ-$MnO_2$ (61.8%) > ε-$MnO_2$ (60.1%) > β-$MnO_2$ (50.1%) respectively. They found that α-$MnO_2$ was the most active catalyst among six kinds of $MnO_2$ for removing both 4-NP and TOC at neutral pH. Similarly, Liang et al. [47] found that the catalytic activity of catalysts increased in the order of β-$MnO_2$ < γ-$MnO_2$ < δ-$MnO_2$ ≈ α-$MnO_2$. Saputra et al. [48] observed the activation of ozone to produce sulfate radicals to degrade phenol, and the catalytic activity analogous order of β-$MnO_2$ < γ-$MnO_2$ < α-$MnO_2$.

Dong et al. [49] reported β-$MnO_2$ as catalyst in the degradation of COD in wastewater. They found that the presence of β-$MnO_2$ nanowires accelerated the removal of COD remarkably (27.1%). Moreover, this experiment showed that β-$MnO_2$ nanowires were beneficial for the adsorption and decomposition of ozone.

Nawaz et al. [42] investigated the catalytic ozonation of 4-nitrophenol (4-NP) using α-$MnO_2$ and found that superoxide radicals, rather than hydroxyl radicals, make a great contribution to the degradation of 4-NP. An ozone molecule attached with α-$MnO_2$ on its active sites detaches with another ozone molecule to produce oxygen and superoxide free radicals [50,51]. Zhao et al. [52] studied the α-$MnO_2$ catalytic ozonation of phenol in water. The results showed that 94.9% of phenol was removed in α-$MnO_2$/$O_3$, which was less than two times higher than phenol removal (52.4%) by single ozonation. Moreover, the *O, ·OH, $O_2{}^-$, and *$O_2$ were not mainly involved in the catalytic ozonation of phenol. Electrons from the surface Mn (III) improved ozone decomposition to some active species and the oxidation of lattice oxygen enhanced the reversion of Mn (IV) to Mn (III). The balance between O (−II)/O (0) and Mn (III)/Mn (IV) was the primary factor for the catalytic performance of $MnO_2$ (Equations (1)–(3)).

$$4Mn(III)O_2 - OH(H_2O) + 4O_3 - 4e \rightarrow 4Mn(IV)O_2 + \text{active oxygen species} \tag{1}$$

$$4Mn(IV) + 2O(-II) \rightarrow 4Mn(III) + O_2 \tag{2}$$

$$2O_3 + 4e \rightarrow 2O(-II) + 2O_2 \tag{3}$$

**Table 3.** Manganese oxide for catalytic ozonation studies.

| Catalyst | Pollutants | Operation Conditions | | Comments | References |
|---|---|---|---|---|---|
| $\delta$-$MnO_2$ | Bisphenol A (BPA) | [Cat] = 0.1 g/L [Pull]$_0$ = 50 mg/L [Time] = 20 min | [$O_3$] = 4 mg/L [%] = 68.2% | The strong interaction among the catalyst surface, ozone and organic molecules rather than hydroxyl radicals were responsible for the degradation of bisphenol A. | Luo et al. (2018) [41] |
| $\alpha$-$MnO_2$ | 4-Nitrophenol (4-NP) | [Cat] = 0.1 g/L [Pull]$_0$ = 50 mg/L [Time] = 30 min | [$O_3$] = 5.0 mg/min pH = 7.0 [%] = 100% | Crystal phase was a vital factor determining the catalytic activity of $MnO_2$. | Nawaz et al. (2017) [46] |
| $MnO_2$ | BPA | [Cat] = 0.05 g/L [Pull]$_0$ = 50 mg/L [Time] = 30 min | [$O_3$] = 2 mg/L [%] = 90% | Two three-dimensional (3D) $MnO_2$ were synthesized and showed the excellent adsorption capacity and catalytic activity. Catalytic ozonation of BPA was dominated by $\bullet O_2^-$ and $\bullet OH$. | Tan et al. (2017) [53] |
| $\alpha$-$MnO_2$ | Phenol | [Cat] = 1.0 g/L [Pull]$_0$ = 300 mg/L [Time] = 60 min | [$O_3$] = 0.80 mg/min pH = 6.4 | The surface hydroxyl groups acted as the active sites in producing active oxygen species, and Lewis acid sites as the reactive centers for catalytic ozonation. | Zhao et al. (2014) [52] |
| $MnO_2$ | Phenol | [Cat] = 0.2 g/L [Pull]$_0$ = 23 mg/L [Time] = 60 min | [$O_3$] = 2.5 mg/min pH = 7.0 [%] = 100% | $MnO_2$ had higher active than $Mn_2O_3$ and $Mn_3O_4$ due to its higher electron transfer ability and higher amount of oxygen defects or surface hydroxyl groups. | Nawaz et al. (2016) [43] |

### 4.2. Titanium Dioxide

Titanium dioxide ($TiO_2$) has been used for a large variety of applications, including the photocatalytic hydrogen production from water, the decomposition and synthesis of organic chemicals, the removal of pollutants from the environment, the reduction of $CO_2$ to chemical fuel, the oxidation of CO, in electron conductors in dye-sensitized solar cells, rechargeable batteries, super-capacitors, sensors, and biomedical devices [54–56]. $TiO_2$ is one of the most widely used and intensely studied materials in photocatalysis. The photoexcitation of semiconductor particles promotes an electron from the valence band to the conduction band to generate a positive hole. Both reductive and oxidative processes can occur at/or near the surface of the photo excited semiconductor particle. Moreover, it is known to be a popular catalyst for ozonation reactions [57,58], due to its low cost, nontoxicity, and high stability in different environments. It can accelerate the ozonation process for the degradation of different kinds of pollutants [59,60] (Table 4).

The $TiO_2/O_3$ process was used for the degradation of nitrobenzene (NB) [58]. It was found that the removal efficiency of NB was significantly improved in the presence of a catalyst, compared with non-catalytic ozonation. The catalytic activity of rutile was better than anatase. Moreover, an increase in the initial NB concentration enhanced NB removal by ozonation and catalytic ozonation.

Rosal et al. [57] studied the use of the $TiO_2$ catalyst for the degradation of naproxen and carbamazepine. Compared with 62% of TOC removal efficiency of naproxen and 73% for carbamazepine that was obtained in pH 5, under the same conditions, 50% of naproxen and carbamazepine were removed by ozonation alone at pH 7. They found that the $TiO_2$ promoted the decomposition of ozone during catalytic ozonation under acidic conditions, while at neutral pH it acted as an inhibitor of the ozone decomposition. The catalyst enhanced the mineralization of naproxen and carbamazepine especially in acidic conditions. In another report [61], the authors used $TiO_2$ catalytic ozonation of clofibric acid from aqueous solution and found similar results. A set of stopped-flow experiments showed that the effect of the catalyst was probably due to the adsorption of organics on catalytic sites rather than the promotion of ozone decomposition.

Song et al. [62] reported $TiO_2$ for the degradation of phenol in water and found that the surface OH groups were responsible for the degradation of phenol. The use of BET, XRD, and TEM analyses showed that $TiO_2$ had high catalytic activities due to their specific surfaces and crystallite phases, whereas the morphology had very little influence on the process. Different crystal phases had different quantities of oxygen vacancy sites, which contributed to the performance of the catalytic ozonation [63–66]. There were a number of surface OH groups due to greater oxygen vacancy sites located in the rutile rather than other crystallite phases of $TiO_2$. However, it was found that the pH of the solution was not controlled in this study. They thought the degradation of phenol relayed strongly on the concentration of surface OH.

**Table 4.** Titanium dioxide for catalytic ozonation studies.

| Catalyst | Pollutants | Operation Conditions | | Comments | References |
|---|---|---|---|---|---|
| TiO$_2$ | Naproxen and carbamazepine | [Cat] = 1.0 g/L<br>[Pull]$_0$ = 15 mg/L<br>[Time] = 20 min<br>Naproxen: [%] = 62%<br>Carbamazepine: [%] = 73% | [O$_3$] = 40 mg/L<br>pH = 5.0 | The adsorption of organics could play a vital role in the reaction. The improvement of mineralization would not merely depend on the production of hydroxyl radicals from ozone. | Rosal et al. (2008) [58] |
| Nano-TiO$_2$ | Mitrobenzene | [Cat] = 0.1 g/L<br>[Pull]$_0$ = 0.06 mg/L<br>[Time] = 20 min | [O$_3$] = 0.367 mg/L<br>pH = 10.0<br>[%] > 60% | The increase of dose of catalyst did not affect the degradation of nitrobenzene. TiO$_2$-catalyzed ozonation followed a radical-type mechanism. | Yang et al. (2007) [59] |
| TiO$_2$ | Ortho-toluidine (OT) | [Cat] = 1.2 g/L<br>[Pull]$_0$ = 50 mg/L<br>[Time] = 60 min | [O$_3$] = 16.6 mg/L<br>pH = 7.0<br>[%] = 96% | The degradation of OT was higher at neutral pH than at alkaline or acidic ones. Degradation of OT followed pseudo-first order kinetics. OT oxidation was occurred through hydroxyl radical mechanism. | Shokri et al. (2017) [5] |
| TiO$_2$ | 4-chloronitrobenzene (4-CNB) | [Cat] = 0.2 mg/L<br>[Pull]$_0$ = 0.1 mg/L<br>[Time] = 20 min | [O$_3$] = 0.35 mg/L<br>pH = 5.3<br>[%] = 64% | The catalyst exhibited the best catalytic activity for removing 4-CNB, which was 9 times higher than ozone alone. Degradation of TiO$_2$ followed a radical-type mechanism. | Ye et al. (2012) [67] |

### 4.3. Iron Oxides

Iron oxide catalysts have been widely used in wastewater treatment (Table 5), like dyes [68], phenol [69], p-chlorobenzoic acid [70], nitrobenzene [71], and the tertiary treatment of industrial wastewater [72]. They have received increasing attention due to their advantages, such as (1) abundance; (2) non-toxicity; (3) and their own special performances—magnetite ($Fe_3O_4$) possesses magnetic properties and iron oxyhydroxide (FeOOH) has a high density of hydroxyl groups; (4) their biocompatibility and recyclability.

An Fe-based catalyst was used as a heterogeneous catalyst for the ozonation of industrial wastewater [73]. The authors found that the total organic carbon (TOC) removal was high, at 78.7%, which was two times higher than the TOC removal efficiency (31.6%) by ozonation alone. In this process, pH was a key operational parameter. Fe-based catalyst could possess a high activity at pH 6–9 [74–76]. On the contrary, at pH 6, many molecular ozone and little $CO_3^{2-}$ existed in the weakly acidic solution, which led to better COD removal.

Wang et al. [77] proposed three prepared-FeOOH catalysts ($SO_4^{2-}$-FeOOH, $Cl^-$-FeOOH, and $NO_3^-$-FeOOH) for improving the ozonation effectiveness of ibuprofen (IBU). Among them, the degradation rate of IBU was reduced following the order $SO_4^{2-}$-FeOOH (40.2%) > $NO_3^-$-FeOOH (35.7%) > $Cl^-$-FeOOH (34.6%). The water pH was a vital factor that determined the charge properties of surface hydroxyl groups at oxide/water interface (Equations (4) and (5)) [78]. From this experiment, they found the $pH_{zpc}$ of $SO_4^{2-}$-FeOOH (7.12) was closer to the pH value of the ibuprofen solution (7.05) than of the other two precursor syntheses. $SO_4^{2-}$-FeOOH had the highest catalytic activity among three kinds of catalysts. Furthermore, $SO_4^{2-}$-FeOOH owned more hydroxyl groups which improved the catalytic activity by forming hydroxyl radical. Sui et al. [72] reported that FeOOH could effectively promote the production of hydroxyl radicals ($\cdot OH$) at pH 4.0 and 7.0, promoting the degradation efficiency of oxalic acid. At pH 4.0, almost all the hydroxyl groups existed in the form of $-OH^{2+}$, and $^-OH$ was the predominant species at pH 7.0. Figure 3 shows the variation of surface hydroxyl species of FeOOH in different solution pH [79]. Both the positive charge state of $Me-OH^{2+}$ (Equations (6)–(8) [80,81]) and neutral state of $Me-OH$ (Equations (9)–(11) [66,81,82]) were beneficial to activate ozone to generate hydroxyl radicals.

$$MeOH_2^+ \Longleftrightarrow MeOH + H^+ \tag{4}$$

$$MeOH + OH^- \Longleftrightarrow MeO^- + H_2O \tag{5}$$

$$O_3 + Me - OH_2^+ \rightarrow Me - OH^+ + HO_3^\cdot \tag{6}$$

$$HO_3^\cdot \rightarrow HO^\cdot + O_2 \tag{7}$$

$$Me - OH^{\cdot+} + H_2O \rightarrow Me - OH_2^+ + HO^\cdot \tag{8}$$

$$2O_3 + Me - OH \rightarrow Me - O_2^{\cdot-} + HO_3^\cdot + O_2 \tag{9}$$

$$HO_3^\cdot \rightarrow HO^\cdot + O_2 \tag{10}$$

$$Me - O_2^{\cdot-} + O_3 + H_2O \rightarrow Me - OH + O_2 + HO_3^\cdot \tag{11}$$

Zhang et al. [83] investigated the relationship between the surface hydroxyl groups of FeOOH and the generation of the hydroxyl radical ($OH\cdot$). They compared with $\alpha$-FeOOH, $\beta$-FeOOH, and $\gamma$-FeOOH and found that not all surface hydroxyl groups possessed high catalytic activity, and the weak surface MeO–H bonds were favorable sites for the generation of $OH\cdot$. More importantly, they found that a stronger electrophilic H and nucleophilic O would facilitate the interaction between the surface hydroxyl group and the dipole molecule of ozone to promote $OH\cdot$ generation due to a weaker surface Me–O bond. Zhang et al. [71] used synthetic goethite (FeOOH) for the degradation of Nitrobenzene (NB). They found that catalytic ozonation with FeOOH could improve NB degradation, which was about two times higher than that in uncatalyzed ozonation. In this experiment, the

protonated and deprotonated surface hydroxyl group were a disadvantage for the production of OH·, and the nearly neutral surface hydroxyl groups were beneficial for producing OH· to improve the ozonation of NB. Figure 4 illustrates the possible pathway of hydroxyl radical generation induced by the surface hydroxyl groups of the FeOOH [71]. They also found that the higher surface hydroxyl densities of FeOOH (0.502 mmolg$^{-1}$) contributed to its activity in the catalytic ozonation of NB. However, Zhang et al. [83] reported that no correlation could be established between the surface hydroxyl densities and the catalytic activities. As for α-FeOOH, β-FeOOH, and γ-FeOOH, their surface hydroxyl densities followed the decreasing order of β-FeOOH > γ-FeOOH > α-FeOOH, but their catalytic activities followed the decreasing order of α-FeOOH > β-FeOOH > γ-FeOOH. There was a relation between the catalytic activity of the oxo-hydroxides and some specific properties of their surface hydroxyl groups.

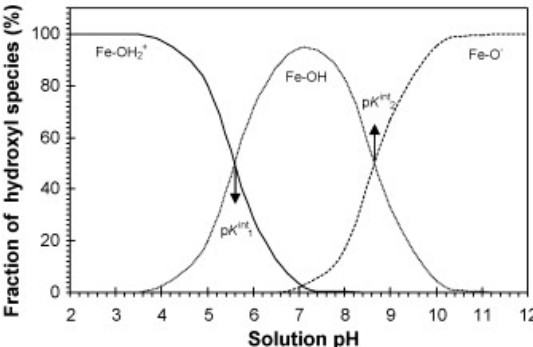

**Figure 3.** Distribution of different surface hydroxyl species on FeOOH as a function of pH.

**Figure 4.** The possible pathway of hydroxyl radical generation.

As reported by Zhu et al. [84], the ordered mesoporous $Fe_3O_4$ catalyst presented high catalytic activity for removing atrazine (ATZ) compared with conventional $Fe_3O_4$ nanoparticles. The ATZ removal with the om-$Fe_3O_4$ catalyst, nano-$Fe_3O_4$, ozonation alone could reach 82.0%, 25.0%, and 9.1%, respectively. The plausible mechanism of ATZ ozonation by ordered mesoporous $Fe_3O_4$ and the most popular reactions could be described in Equations (12)–(18). The redox cycles of $Fe^{2+}/Fe^{3+}$ were responsible for hydroxyl radical (OH·) generation.

$$Fe^{2+} + O_3 \rightarrow FeO^{2+} + O_2 \tag{12}$$

$$FeO^{2+} + H_2O \rightarrow Fe^{3+} + \cdot OH + OH^- \tag{13}$$

$$Fe^{3+} + O_3 + H_2O \rightarrow FeO^{2+} + \cdot OH + O_2 + H^+ \tag{14}$$

$$\cdot OH + ATZ \rightarrow [\cdots \text{many steps}] \rightarrow CO_2 + H_2O \tag{15}$$

$$O_3 + ATZ \rightarrow [\cdots \text{many steps}] \rightarrow CO_2 + H_2O \tag{16}$$

$$Fe^{3+} + O_2^{\cdot-} \rightarrow Fe^{2+} + O_2 \tag{17}$$

$$Fe^{3+} + HO_2^{\cdot-} \rightarrow Fe^{2+} + H^+ + O_2 \tag{18}$$

**Table 5.** Iron oxides for catalytic ozonation studies.

| Catalyst | Pollutants | Operation Conditions | | Comments | References |
|---|---|---|---|---|---|
| $\beta$-FeOOH | 4-chlorophenol (4-CP) | [Cat] = 0.1 g/L<br>[Pull]$_0$ = 2 $\times$ 10$^{-3}$ mol/L<br>[Time] = 40 min | [O$_3$] = 0.6 mg/min<br>pH = 3.5<br>[%] = 99% | The catalytic ability of the $\beta$-FeOOH during ozonation process was found to be shown at lower pH. | Oputu et al. (2015) [74] |
| FeOOH | Oxalic acid | [Cat] = 2.0 g/L<br>[Pull]$_0$ = 1 $\times$ 10$^{-5}$ mol/L<br>[Time] = 30 min | [O$_3$] = 0.45 mg/min<br>pH = 7.0<br>[%] = 54% | FeOOH could effectively improve the generation of hydroxyl radicals ($\cdot$OH). Hydroxyl groups in different situations such as neutral state and positive charge state could act as the active sites for the decomposition of ozone and the generation of hydroxyl radicals. | Sui et al. (2010) [79] |
| $\alpha$-FeOOH | Nitrobenzene | [Cat] = 0.2 g/L<br>[Pull]$_0$ = 2.96 $\times$ 10$^{-6}$ mol/L<br>[Time] = 15 min | [O$_3$] = 1.2 mg/L<br>pH = 7.3 | The neutral surface hydroxyl species of $\alpha$-FeOOH had high activity to catalyze $\cdot$OH generation from aqueous ozone due to the surface OH-ozone interaction. | Zhang al. (2008) [83] |
| Fe$_3$O$_4$ | Atrazine (ATZ) | [Cat] = 0.2 g/L<br>[Pull]$_0$ = 5.0 $\times$ 10$^{-6}$ mol/L<br>[Time] = 2 min | [O$_3$] = 4.8 mg/L<br>pH = 9.5<br>[%] = 95% | The redox cycles of Fe$^{2+}$/Fe$^{3+}$ benefited the generation of $\cdot$OH. Mesoporous Fe$_3$O$_4$ presented low iron leaching, good stability and easy to separate. | Zhu et al. (2017) [84] |
| Micro-size Fe$^0$ (mFe$^0$) | P-nitrophenol (PNP) | [Cat] = 40 g/L<br>[Pull]$_0$ = 3.6 $\times$ 10$^{-3}$ mol/L<br>[Time] = 60 min | [O$_3$] = 7.6 mg/L<br>pH = 5.3<br>[%] = 89.5% | High degradation of PNP in aqueous solution was due to the synergetic effect between O$_3$ and Fe$^0$. | Xiong et al. (2016) [85] |

### 4.4. Aluminum Oxides

Alumina-based catalysts have been studied in the catalytic ozonation of different organic contaminants because of their high catalytic activities (Table 6). As reported in several studies, aluminum oxides might enhance the generation of ·OH.

Aghaeinejad-Meybodi et al. [86] reported that $\gamma$-Al$_2$O$_3$ dramatically improved the removal rate of Fluoxetine, up to 96.14%. The removal efficiency enhanced when the molar ratio of O$_3$ to Fluoxetine decreased (13 to 4) while the efficiency declined with a further decrease of the molar ratio. They found that the effective parameters of the removal efficiency of Fluoxetine with both ANN and CCD models followed this order: Initial Fluoxetine concentration < nano-$\gamma$-Al$_2$O$_3$ catalyst dosage < ozone concentration < reaction time. Keykavoos et al. [87] studied the catalytic activity of alumina for the ozonation of bisphenol A (BPA). They found that the degradation efficiency for BPA was higher in catalytic ozonation (90%) as compared to ozonation alone (35%). The presence of a catalyst in the ozonation system was more effective than the combination of ozonation and catalyst as an adsorbent for ultimate by-products.

Vittene et al. [88] used $\gamma$-Al$_2$O$_3$ as a catalyst for the degradation of 2,4-Dimethylphenol (2,4-DMP) without pH adjustment. The degradation efficiency for 2,4-DMP was found to be more than four times higher than without a catalyst. Three carboxylic acids were mainly formed during 2,4-DMP ozonation in the following decreasing amount: Oxalic acid < formic acid < acetic acid. $\gamma$-Al$_2$O$_3$ was an amphoteric solid with Lewis acid AlOH (H$^+$) sites and basic Al-OH sites (Figure 5) [18]. A part of the carboxylic acid was adsorbed on the basic sites of $\gamma$-Al$_2$O$_3$, resulting in a decrease of catalytic activity. They also found that sodium ions could prevent carboxylates adsorption on $\gamma$-Al$_2$O$_3$.

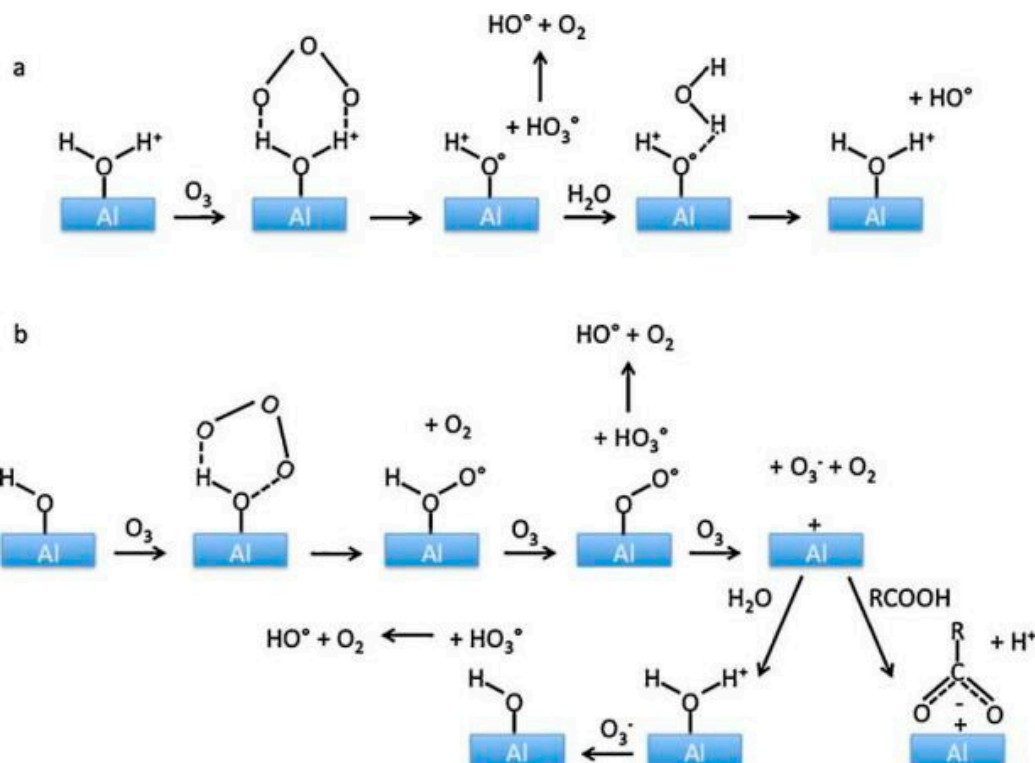

**Figure 5.** Expected interaction of O$_3$ with (**a**) acid and (**b**) basic sites of $\gamma$-Al$_2$O$_3$. (**b**) Hypothesis for adsorption mechanism of carboxylic acids with the basic sites of $\gamma$-Al$_2$O$_3$ during ozonation.

Qi et al. [88] used γ-AlOOH (HAO) and γ-Al$_2$O$_3$ (RAO) for the degradation of 2-isopropyl-3-meth-oxypyrazine (IPMP). In neutral water pH, the removal efficiency of IPMP in the presence of HAO and RAO were over 90.0%. The catalytic activities of both HAO and RAO were insignificant under lower pH. They found that OH· produced in water with the presence of HAO and OH· mainly generated around the surface of RAO. HAO and RAO presented different reaction mechanisms [88] (Figure 6). In another report, Qi et al. [89] also studied the effect of inorganic ions on ozone decomposition catalyzed by the aluminum oxides that was related to hydroxyl groups on the surfaces of the catalysts.

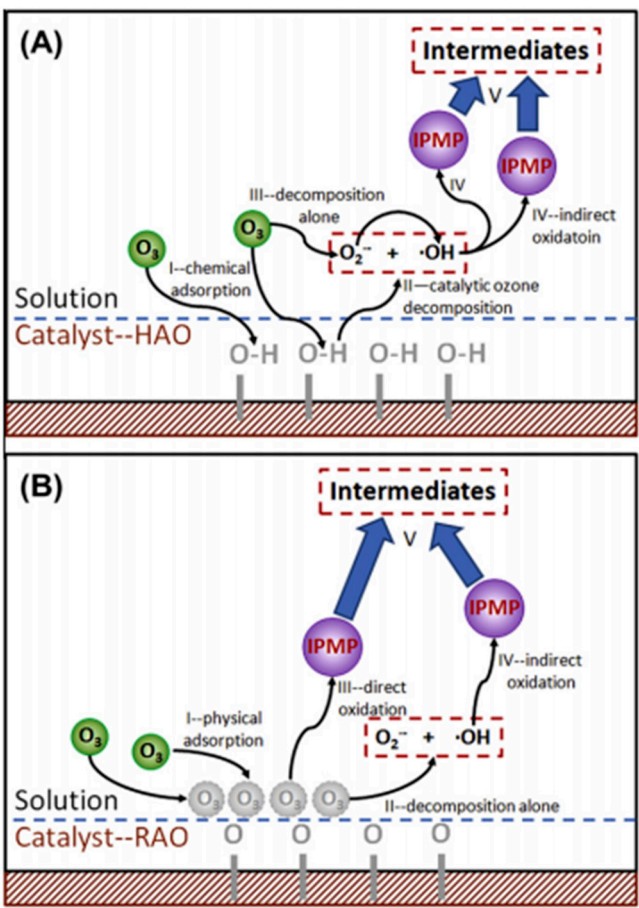

**Figure 6.** Proposed reaction pathway in catalytic ozonation by γ-AlOOH (**A**) or γ-Al$_2$O$_3$ (**B**).

**Table 6.** Aluminum oxides for catalytic ozonation studies.

| Catalyst | Pollutants | Operation Conditions | | Comments | References |
|---|---|---|---|---|---|
| Alumina | Bisphenol A (BPA) | [Cat] = 1.0 g/L [Pull]$_0$ = 10 mg/L [Time] = 60 min | [O$_3$] = 4.5 mg/L pH = 5.0 [%] = 90% | The increase of alumina catalyst dose from 0.5 g/L to 4 g/L did not exhibit a big effect on the TOC removal. | Keykavoos et al. (2013) [87] |
| γ-Al$_2$O$_3$ | 2,4-Dimethylphenol (2,4-DMP) | [Cat] = 2.0 g/L [Pull]$_0$ = 50 mg/L [Time] = 22 min | [O$_3$] = 2 × 10$^{-6}$ mg/L pH = 4.5 [%] = 100% | It was found that the reaction followed hydroxyl radical mechanism. No adsorption of 2,4-DMP occurred on γ-Al$_2$O$_3$. | Vittenet et al. (2015) [18] |
| γ-AlOOH (HAO) and γ-Al$_2$O$_3$ (RAO) | 2-isopropyl-3-meth-oxypyrazine (IPMP) | [Cat] = 0.5 g/L [Pull]$_0$ = 0.04 mg/L [Time] = 10 min HAO: [%] = 94.2% RAO: [%] = 90.0% | [O$_3$] = 0.5 mg/L pH = 7.05 | Both HAO and RAO showed the good stability with low aluminum ions leaching. Surface hydroxyl groups were important reaction sites for HAO but not for RAO. | Qi et al. (2013) [88] |
| Nano-perfluorooctyl alumina (PFOAL) | Tert-butyl ether (MTBE) | [Cat] = 5.0 g/L [Pull]$_0$ = 1.0 g/L [Time] = 90 min | pH = 6.7 [%] = 98.18% | Degradation using PFOAL catalyst was about two times higher than single ozonation. The adsorption of MTBE on PFOAL followed pseudo-second-order kinetics. | Kiadehi et al. (2017) [90] |
| γ-AlOOH | 2,4, 6-trichloroanisole (TCA) | [Cat] = 0.2 g/L [Pull]$_0$ = 0.025 mg/L [Time] = 10 min | [O$_3$] = 0.5 mg/L pH = 6.0 [%] = 80% | The significant decrease in the TCA degradation rate due to the transformation of the crystal phase and the reduction of surface hydroxyl groups. | Qi et al. (2009) [91] |

### 4.5. Magnesium Oxide

MgO has been given considerable attention as it has shown good catalytic ozonation performances for the degradation of several types of compounds including phenol [92], 4-chlorophenol [93], formaldehyde [94], and dyes [95] due to its unique features, such as a highly efficient activity and reactivity, a high structural stability in many surface basic sites [96], destructive adsorbance [97], a high specific surface area [98], environment friendliness, low toxicity [92,98–101], and it is hard and almost non-soluble in water. The catalytic ozonation studies are summarized in Table 7.

Zhu et al. [102] studied the catalytic activity of the nano-MgO catalyst for the degradation of quinoline present in the biologically pretreated coal gasification wastewater. Under the same conditions, they found that 90.7% of quinoline was removed after 15 min. It was found that the hydroxyl radical generated due to decomposition of ozone on the nano-MgO catalyst was responsible for the degradation and mineralization of quinoline, thereby, improving degradation efficiency for quinoline. The results showed the small adsorption could be neglected. Detailed proposed mechanisms in this experiment are revealed in Figure 7 [102].

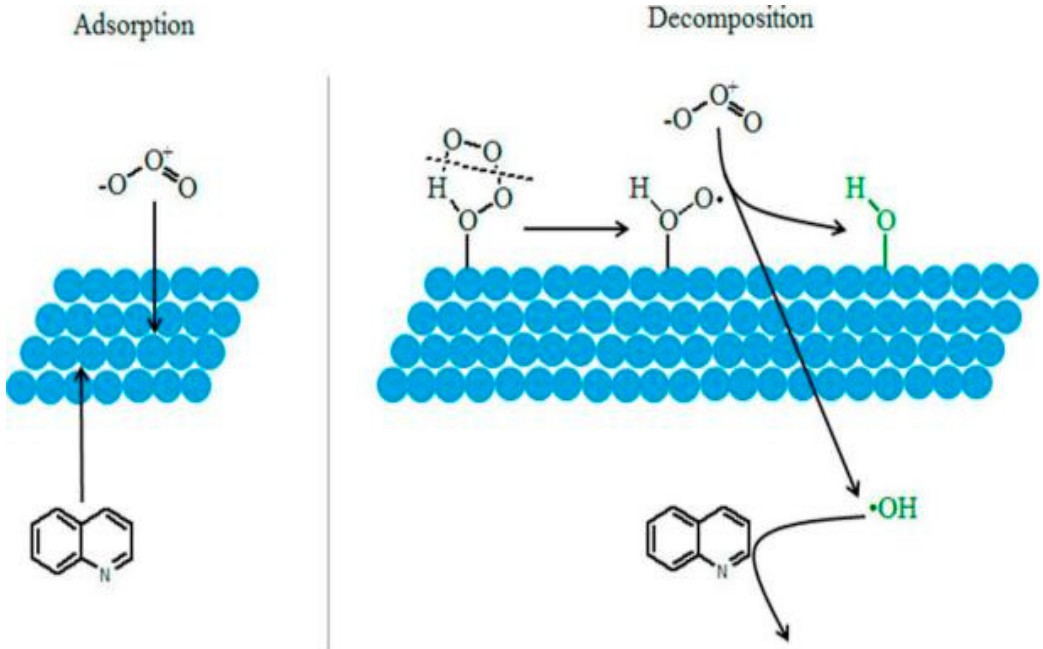

**Figure 7.** Mechanism of quinoline degradation in catalytic ozonation.

Mashayekh-Salehi et al. [103] prepared MgO powder via a novel thermal/sol-gel method and used it for enhancing the ozonation effectiveness of acetaminophen. They found that MgO had a high catalytic activity. The basic surface functional groups played a vital role in promoting the transformation of ozone to OH·. With the addition of MgO, the reaction between the $O_3$ molecules and both the O and H atoms of the hydroxyl groups existed on the surface of MgO. The reaction between OH· and ACT molecules occurred in the bulk solution. When releasing OH· to the solution, MgO adsorbed $H_2O$ molecules and dissociated them into OH· and $H^+$, which enhanced the OH· generation from water [104,105]. A mechanism for degradation of ACT was proposed and given in Equations (19)–(26).

- Direct oxidation with $O_3$ molecules on MgO surface:

$$MgO^{-O_3} + ACT \rightarrow CO_2 + H_2O + \text{intermediates} \tag{19}$$

$$MgO^{-ACT} + O_3 \rightarrow CO_2 + H_2O + \text{intermediates} \tag{20}$$

- Radical type catalytic oxidation on MgO surface:

$$MgO - s + O_3 \rightarrow MgO - S^{O\equiv O-O} \rightarrow MgO^{-S^{O\cdot}} + O_2 \tag{21}$$

$$MgO^{-S^{O\cdot}} + 2H_2O + O_3 \rightarrow MgO - S^{(\cdot OH)_2} + 3\cdot OH + 2O_2 \tag{22}$$

$$MgO - S^{(\cdot OH)_2} + ACT \rightarrow CO_2 + H_2O + \text{intermediates} \tag{23}$$

$$MgO^{-ACT} + \cdot OH \rightarrow CO_2 + H_2O + \text{intermediates} \tag{24}$$

- Direct oxidation with $O_3$ molecules in the bulk solution:

$$O_3 + ACT \rightarrow CO_2 + H_2O + \text{intermediates} \tag{25}$$

- Radical type catalytic oxidation in the bulk solution:

$$\cdot OH + ACT \rightarrow CO_2 + H_2O + \text{intermediates} \tag{26}$$

Chen et al. [93] synthesized three different types of crystal facet of (111), (110), and (200) of MgO and compared the catalytic capabilities among them by the catalytic ozonation of 4-chlorophenol. It was found that different dosages generated the same surface area of MgO, their catalytic capabilities were in the following order: MgO (200) < MgO (110) < MgO (111). This indicated that crystal facet played an important role in catalytic activity. Because of the edges, steps, and kinks, each facet increased in the order of MgO (200) < MgO (110) < MgO (111). The use of radical scavengers tert-butanol (TBA) showed that the hydroxyl radicals were responsible for the degradation of 4-chlorophenol.

Moussavi et al. [92] reported MgO for the COD removal and degradation of phenol from saline wastewater, particularly those containing inhibitory and toxic compounds and studied the influence of several variables, such as the dose of MgO, pH, and NaCl concentration on the efficiency of the catalytic process. The concentration of NaCl had no adverse effect on the phenol degradation. At neutral pH, they found that 70% of the COD and 96% of the phenol were removed in the catalytic ozonation process (COP) and 39% of a synergistic influence for phenol degradation in the COP. In other report [106], they also used MgO for the degradation of reactive red 198 azo dye. With addition of MgO catalyst, the rate of RR198 degradation greatly increased, thereby reducing the reaction time compared to single ozonation. At pH 8, the reaction time of the complete removal of color as short as 9 min, while the time at ozonation alone was 30 min.

**Table 7.** Magnesium oxide for catalytic ozonation studies.

| Catalyst | Pollutants | Operation Conditions | | Comments | References |
|---|---|---|---|---|---|
| MgO | 4-chlorophenol | [Cat] = 1.0 g/L<br>[Pull]$_0$ = 100 mg/L<br>[Time] = 30 min | [O$_3$] =2.5 mg/min<br>pH = 6.2<br>[%] = 99.3% | The pseudo first-order reaction constant of 4-chlorophenol removal in catalytic ozonation using MgO (111), MgO (110), MgO (200) catalyst mixed with ozone were 4.8, 2.5, 3.1 times higher than that of ozonation alone, respectively. | Chen et al.<br>(2015)<br>[93] |
| Nano-MgO | Quinoline | [Cat] = 0.2 g/L<br>[Pull]$_0$ = 20 mg/L<br>[Time] = 15 min | [O$_3$] =2.0 mg/L<br>pH = 6.8<br>[%] = 90.7% | Nano-MgO could accelerate ozone decomposition and behave via a hydroxyl radical mechanism. The catalyst remained stable catalytic ability. | Zhu et al.<br>(2017)<br>[102] |
| MgO | Acetaminophen (ACT) | [Cat] = 2.0 g/L<br>[Pull]$_0$ = 50 mg/L<br>[Time] = 15 min | [O$_3$] = 1.8 mg/min<br>pH = 5.4<br>[%] = 100% | Kinetics of ACT oxidation showed that the rate of degradation and mineralization ACT was 18.8 times and 7.8 times in the MgO/O$_3$ process compared to the ozonation alone. Degradation of ACT was governed by hydroxyl radical mechanism. | Mashayekh-Salehi et al.<br>(2017)<br>[103] |
| MgO | 2,4-Dichlorophenol (2,4-DCP) | [Cat] = 0.3 mg/L<br>[Pull]$_0$ = 50 mg/L<br>[Time] = 50 min | pH > 7.0<br>[%] = 99.99% | Effect of operational parameters like solution pH, ozonation time, dose of MgO and initial 2,4-DCP concentration. | Mohammadi et al.<br>(2017)<br>[107] |
| Nano-MgO | Metronidazole (MNZ) | [Cat] = 0.25 g/L<br>[Pull]$_0$ = 40 mg/L<br>[Time] = 20 min | [O$_3$] = 8.3 mg/min<br>pH = 10.0<br>[%] = 93.5% | The introduction of MgO nanocrystals contributed to the increase of MNZ removal and the decrease of required time compared to the conventional ozonation. | Kermani et al.<br>(2015)<br>[108] |
| MgO(111) | Nitrobenzene | [Cat] = 1.0 g/L<br>[Pull]$_0$ = 50 mg/L<br>[Time] = 30 min | [O$_3$] = 5.0 mg/L<br>pH = 12.0<br>[%] = 95.7% | Catalytic activities of three catalyst followed MgO (111) > CP-MgO > MnO$_x$ for nitrobenzene mineralizaton. MgO (111) had amount of surface O$_2^-$ Lewis basic sites. | Chen et al.<br>(2014)<br>[109] |

### 4.6. Bimetallic Oxides

An increasing number of bimetallic catalysts are becoming popular in heterogeneous catalytic ozonation (Table 8), as bimetallic oxides usually present high catalytic activity and stability compared with monometallic oxides [110,111]. Spinel type oxides act as $AB_2O_4$ (where A and B are metal ions), and have received increasing attention due to their availability, low-cost, structure, valence states, morphologies, and surface defects. Such properties have illustrated that their good catalytic abilities could be competitive with noble metals [112].

Xu et al. [113] prepared $CuAl_2O_4$ and used it for the catalytic ozonation of dye solution. They found that near 100% of color and 87.2% of COD removal of the dye solution were removed in mixed oxide/$O_3$ process within 25 min at neutral pH due to the better textural properties and a higher density of active sites. It was worth mentioning that surface adsorbed $HO\cdot$ and $O_2^{\bullet-}$ played a vital role in the organic's degradation. They also measured the reaction rate value of $CuAl_2O_4/O_3$, $CuO/O_3$ and $Al_2O_3/O_3$, 0.112 min$^{-1}$, 0.071 min$^{-1}$, and 0.074 min$^{-1}$ respectively. Notably, the reaction rate value of $CuAl_2O_4/O_3$ was higher than others. The superior catalytic activity of $CuAl_2O_4$ due to the synergistic effect of active sites on the surface $\equiv Cu^{2+}$ and $\equiv Al^{3+}$. Moreover, they proposed a possible mechanism of radical generation for the catalytic ozonation process over $CuAl_2O_4$ (Equations (27)–(39)).

$$\equiv Cu^{2+} + H_2O \rightarrow \equiv Cu^{2+} - OH + H^+ \tag{27}$$

$$\equiv Cu^{2+} - OH + O_3 \rightarrow \equiv Cu^{2+} - O_3^{\cdot} + OH^{\cdot} \tag{28}$$

$$\equiv Cu^{2+} - O_3^{\cdot} + OH^- \rightarrow \equiv Cu^+ + HO_2^{\cdot} + O_2 \tag{29}$$

$$HO_2^{\cdot} \rightarrow H^+ + O_2^{\cdot-} \tag{30}$$

$$\equiv Cu^+ + O_3 \rightarrow \equiv Cu^{2+} - O_3^{\cdot} \tag{31}$$

$$\equiv Cu^{2+} - O_3^{\cdot} + H^+ \rightarrow \equiv Cu^{2+} + OH^{\cdot} + O_2 \tag{32}$$

$$\equiv Al^{3+} + H_2O \rightarrow \equiv Al^{3+} - OH + H^+ \tag{33}$$

$$\equiv Al^{3+} - OH + O_3 \rightarrow \equiv Al^{3+} - HO_2^{\cdot} + O_2 \tag{34}$$

$$\equiv Al^{3+} - HO_2^{\cdot} + O_3 \rightarrow \equiv Al^{3+} - HO_3^{\cdot} + O_2 \tag{35}$$

$$\equiv Al^{3+} - HO_3^{\cdot} \rightarrow \equiv Al^{3+} + HO^{\cdot} + O_2 \tag{36}$$

$$\equiv Al^{3+} - OH + O_3 \rightarrow \equiv Al^{3+} - O + OH^- + O_2 \tag{37}$$

$$\equiv Al^{3+} - O + H_2O \rightarrow \equiv Al^{3+} + 2HO^{\cdot} \tag{38}$$

$$\equiv Al^{3+} - O + O_3 \rightarrow \equiv Al^{3+} + O_2^{\cdot-} + O_2 \tag{39}$$

Liu et al. [114] studied the catalytic ozonation of dimethyl phthalate (DMP) using Cu-Fe-O nanoparticles (CFO NPs) as catalyst. They found the great DMP removal efficiency was achieved in a wide range of pH from 3 to 9 and the amount of $OH\cdot$ was much higher than that of non-catalytic ozonation at pH 5.7. From this study, we known that the $OH\cdot$ played an important role in removing DMP. It was found that the potential of $\equiv Fe$ (III)/$\equiv Fe$ (II) and $\equiv Cu$ (II)/$\equiv Cu$ (I) cycles greatly influenced the production of $OH\cdot$, but which was found to be negligible. They also proposed the reaction mechanisms for $O_3$/CFO NPs catalytic ozonation (Equations (40)–(49)).

$$\equiv Cu(I)_2O - OH - + O_3 \rightarrow \equiv Cu(I)_2O - OH - O_3 \tag{40}$$

$$O_2 + 4e \rightarrow 2O^{2-} \tag{41}$$

$$O^{2-} + \equiv Cu(I)_2O - OH - O_3 \rightarrow 2 \equiv Cu(II)_2O + HO_2^{\cdot-} \tag{42}$$

$$O_3 + HO_2^{\cdot -} \rightarrow OH^{\cdot -} + O_2^{\cdot -} + O_2 \tag{43}$$

$$\equiv Cu(II) + O_2^{\cdot -} \rightarrow \equiv Cu(I) + O_2 \tag{44}$$

$$\equiv Fe(III) + O_2^{\cdot -} \rightarrow \equiv Fe(II) + O_2 \tag{45}$$

$$\equiv Cu(I) + \equiv Fe(III) \rightarrow \equiv Cu(II) + \equiv Fe(II) \tag{46}$$

$$\equiv Fe(II) - OH - + O_3 \rightarrow \equiv Fe(II) - OH - O_3 \tag{47}$$

$$O^{2-} + 2 \equiv Fe(II) - OH - O_3 \rightarrow \equiv Fe(II)_2 O_3 + HO_2^{\cdot -} + O_2^{\cdot -} \tag{48}$$

$$\cdot OH + CH_3 - R \rightarrow \cdot CH_3 + R - OH \tag{49}$$

Liu et al. [115] successfully prepared four kinds of magnetic spinel ferrite, such as $CoFe_2O_4$, $CuFe_2O_4$, $NiFe_2O_4$, and $ZnFe_2O_4$ and applied them to treat the practical shale gas produced water in catalytic ozonation. From the results, they found that the magnetic spinel ferrite promoted the B [Spsbackslash] C ratio from less than 0.1 to over 0.3 and enhanced the AOC concentration to 5 times. The catalytic performances followed the order of $CuFe_2O_4 > NiFe_2O_4 > CoFe_2O_4 > ZnFe_2O_4$. By Scatchard model and Weber–Morris model, they found that the catalytic capacity depended on surface property. Zhang et al. [116] also synthesized four kinds of magnetic spinel ferrite and studied the catalytic activities of them for ozonation of oxalic acid. They found that the $CoFe_2O_4$ had higher catalytic performance than others ($CoFe_2O_4 > NiFe_2O_4 > CuFe_2O_4 > ZnFe_2O_4$) in catalyzing oxalic acid mineralization, 68.3% of TOC was removed within 120 min. The results showed that these spinel ferrites owned the reducibility and the electron donating capacity. Moreover, they proposed the radical-based mechanistic pathways, which included the interaction of surface hydroxyl groups and surface metal ions with ozone. The removal efficiency of N, N-dimethylacetamide (DMAC) using $CuFe_2O_4$ as catalyst in catalytic ozonation was reported by Zhang et al. [117]. They found that 95.4% of DMAC, 30.1% of COD, and 22.3% of TOC were removed in $CuFe_2O_4/O_3$ process after 120 min. The reaction mechanism of the catalytic ozonation consisted of three different parts, such as heterogeneous catalytic ozonation, homogeneous catalytic ozonation, ozonation alone [117] (Figure 8).

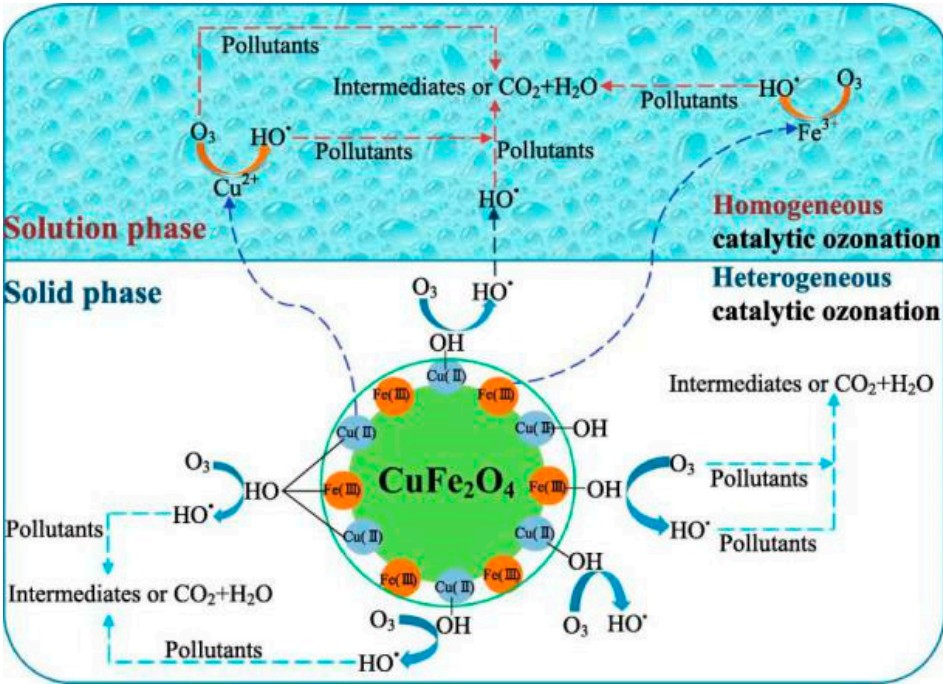

**Figure 8.** Possible reaction mechanism of the $CuFe_2O_4/O_3$ process.

**Table 8.** Bimetallic oxides for catalytic ozonation studies.

| Catalyst | Pollutants | Operation Conditions | | Comments | References |
|---|---|---|---|---|---|
| $Ce_{0.1}Fe_{0.9}OOH$ | Sulfamethazine (SMT) | [Cat] = 0.4 g/L<br>$[Pull]_0$ = 10 mg/L<br>[Time] = 10 min | $[O_3]$ = 20 mg/L<br>pH = 7.0<br>[%] = 100% | $Ce_{0.1}Fe_{0.9}OOH$ was prepared by isomorphous substitution method. The catalyst enhanced the mineralization efficiency of SMT depending on the dosage of ozone and catalyst. | Bai et al. (2016) [118] |
| $ZnAl_2O_4$ | 5-Sulfosalicylic acid (SSal) | [Cat] = 0.2 g/L<br>$[Pull]_0$ = 500 mg/L<br>[Time] = 60 min | $[O_3]$ = 5.0 mg/min<br>pH = 7.0<br>[%] = 64.8% | The $ZnAl_2O_4$ catalyst was prepared by hydrothermal, sol-gel, and coprecipitation methods were compared. Catalyst prepared by hydrothermal method showed better catalytic activity in ozonation. Degradation of SSal followed radical mechanism. | Dai et al. (2018) [119] |
| $ZnAl_2O_4$ | Phenol | [Cat] = 1.0 g/L<br>$[Pull]_0$ = 300 mg/L<br>[Time] = 60 min | $[O_3]$ = 0.75 mg/min<br>pH = 6.4<br>[%] = 73.4% | After using 4 times of catalyst, the removal rate of phenol slightly decreased by 5.7%. Hydroxyl radicals reacted with phenol in bulk solution. $ZnAl_2O_4$ was applied in a wide pH range from 3.3 to 9.3. | Zhao et al. (2016) [120] |
| $MgFe_2O_4$ | Acid Orange II (AOII) | [Cat] = 0.1 g/L<br>$[Pull]_0$ = 50 mg/L<br>[Time] = 40 min | $[O_3]$ = 5.0 mg/L<br>pH = 6.5<br>[%] = 94.1% | $MgFe_2O_4$ had the most catalytic activity among MgO, $Fe_2O_3$ and $MgO+Fe_2O_3$ and possessed a reaction rate constant at least 2.3 times compared to that of $NiFe_2O_4$, $MnFe_2O_4$ and $CuFe_2O_4$. | Lu et al. (2015) [121] |
| $CaMn_3O_6$ and $CaMn_4O_8$ | 4-nitrophenol | [Cat] = 0.1 g/L<br>$[Pull]_0$ = 50 mg/L<br>$CaMn_3O_6$: [Time] = 45 min<br>$CaMn_4O_8$: [Time] = 30 min | $[O_3]$ = 50 mg/L<br>pH = 5.7<br><br>[%] = 100% | The superoxide radicals and singlet oxygen other than hydroxyl radicals were responsible for the degradation and mineralization of 4-nitrophenol. The $CaMn_3O_6$ and $CaMn_4O_8$ exhibited much higher catalytic activities and stabilities than manganese oxides. | Wang et al. (2015) [122] |
| Mn-Ce-O | Antipyrine | [Cat] = 0.1 g/L<br>$[Pull]_0$ = 40 mg/L<br>[Time] = 2 min | $[O_3]$ = 20 mg/L<br>pH = 6.5<br>[%] = 100% | Catalytic ozonaion was governed by hydroxyl radical mechanism. A strengthen of the contribution of surface reactions with a decrease of pH. | Xing et al. (2015) [123] |
| $NiFe_2O_4$ | Phenol | [Cat] = 1.0 g/L<br>$[Pull]_0$ = 300 mg/L<br>[Time] = 60 min | $[O_3]$ = 0.75 mg/min<br>pH = 6.5<br>[%] = 97.6% | The $NiFe_2O_4$-H and $NiFe_2O_4$-C were prepared by hydrothermal and calcined treatments, respectively. Presence of $NiFe_2O_4$-H promoted the degradation of phenol, but $NiFe_2O_4$-C was noneffective. Lewis acid sites were behaved as reactive centers for catalytic ozonation. | Zhao et al. (2013) [124] |
| Fe-Cu-O | Acid Red B (ARB) | [Cat] = 1.0 g/L<br>$[Pull]_0$ = 100 mg/L<br>[Time] = 10 min | $[O_3]$ = 30 mg/min<br>pH = 6.8<br>[%] = 98% | The Fe-Cu-O indicated good stability after four successive recycles. Degradation of ARB followed pseudo-first-order rate equation. | Liu et al. (2013) [125] |
| $SrTiO_3$ | OA | [Cat] = 1.25 g/L<br>$[Pull]_0$ = 100 mg/L<br>[Time] = 60 min | $[O_3]$ = 18.4 mg/L<br>pH = 3.0<br>[%] = 45.8% | Catalyst indicated good stability and efficiency after four successive cycles. | Wu et al. (2011) [126] |

Bai et al. [118] used $Ce_{0.1}Fe_{0.9}OOH$ as catalyst in the ozonation of sulfamethazine (SMT) and found that acidic and neutral pH were responsible for the SMT mineralization. More importantly, they presented the possible reaction pathways for the oxidation of SMT by ozone mixed $Ce_{0.1}Fe_{0.9}OOH$ [118] (Figure 9).

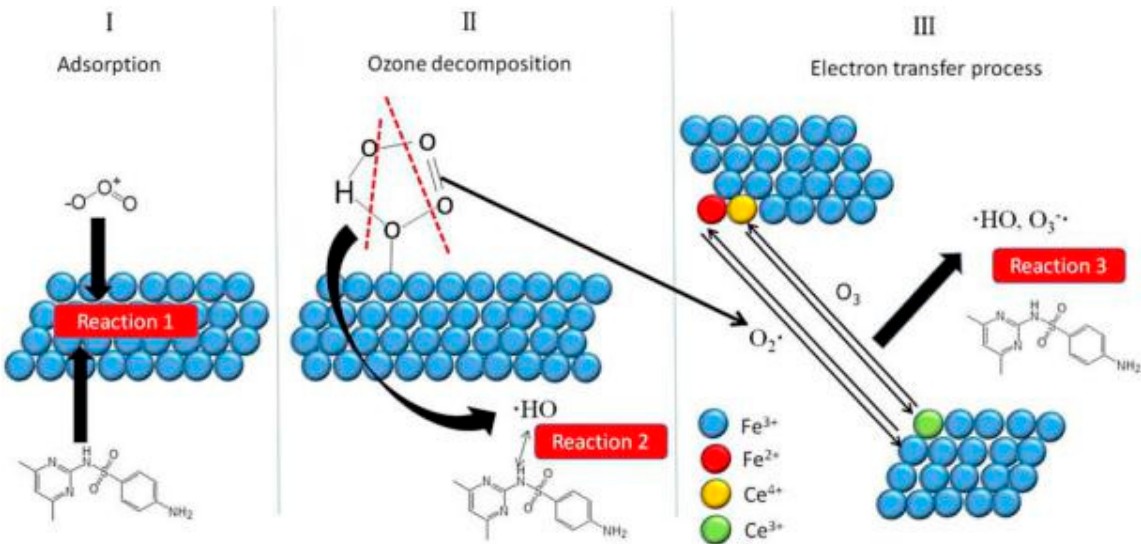

**Figure 9.** Schematic of possible catalytic mechanisms.

## 5. Metal or Metal Oxides on Supports

Metal oxides are loaded on supports, and they act as catalysts for promoting the degradation of organic compounds due to the increase of the surface area and active sites of catalysts [104,127,128]. They have a high catalytic performance in catalytic ozonation processes to remove contaminants in wastewater (Table 9).

Sun et al. [129] investigated the oxidation of clofibric acid (CA) by ozone mixed $MnO_x$/SBA-15. Compared with single ozonation, the removal efficiency of CA was scarcely enhanced, while the removal of TOC was surprisingly improved by catalytic ozonation. It was worth mentioning that the adsorption of CA and TOC on $MnO_x$/SBA-15 were found to be negligible. In $O_3$/$MnO_x$/SBA-15 process, they found that Mn–$OH^{2+}$ acted as the main catalytic sites, where more ·OH were formed and transferred into the aqueous solution. They also [35] studied the $MnO_x$/SBA-15 catalytic ozonation of oxalic acid (OA) and found that $MnO_x$/SBA-15 had a high catalytic ozonation capacity and adsorption ability for OA. The protonated surface Mn-$OH^{2+}$ greatly affected OA adsorption and ·OH initiation. This study followed the mechanism of OH· oxidation. Higher multi-valent $MnO_x$ (Mn (III)/Mn (IV)) benefited OA removal due to the promotion of electron transfer. The probable reaction pathway was proposed in Figure 10 for the ozonation of OA using $MnO_x$/SBA-15 as catalyst [35]. Huang et al. [130] reported that the $MnO_x$/sewage sludge-derived activated carbon ($MnO_x$/SAC) exhibited the strongest catalytic activity at pH 3.5, while inhibited the oxalic acid removal at alkaline pH. Through the addition of hydroxyl radical scavenger and the analysis of the surface composition of catalyst, the results showed that the reaction mechanism involved two parts, such as surface reactions and reactions between the organic matters and hydroxyl radicals in the bulk water, but dominantly surface reactions [130] (Figure 11).

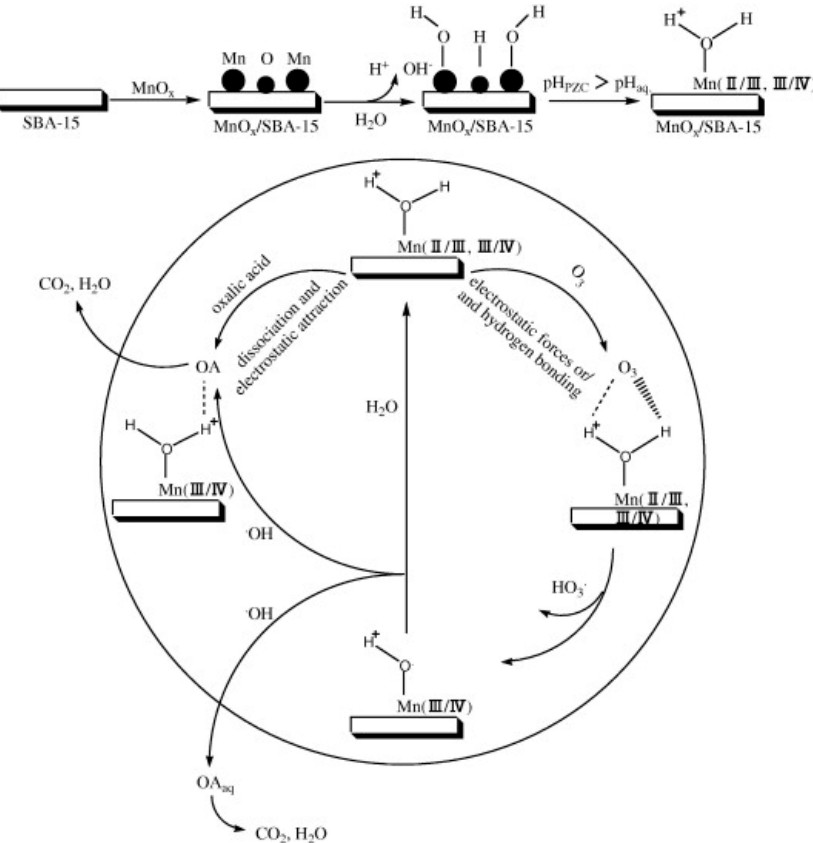

**Figure 10.** Proposed reaction route for oxalic acid (OA) removal in $O_3/MnO_x/SBA-15$.

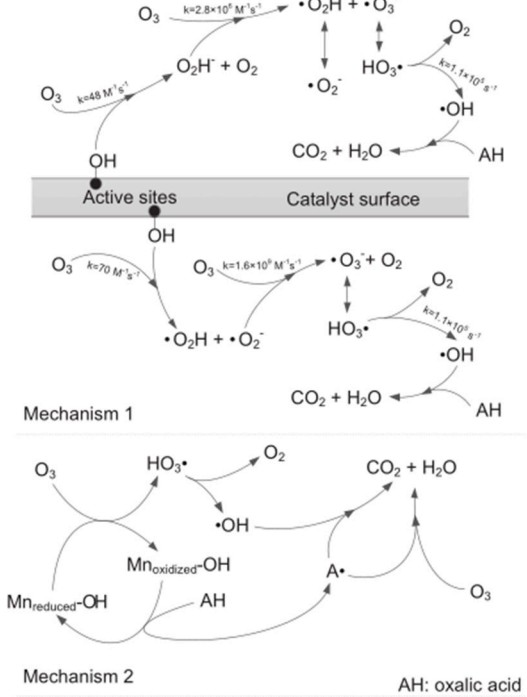

**Figure 11.** Mechanism of heterogeneous catalytic ozonation with $MnO_x/SAC$.

Li et al. [131] synthesized mesoporous MCM-48 and Ce loaded MCM-48 (Ce/MCM-48) to improve the ozonation effectiveness of clofibric acid in aqueous solution. They compared the catalytic capability among Ce/MCM-48, Ce/MCM-41 and MCM-48 by catalytic ozonation of Total Organic Carbon (TOC) and found that the removal efficiency followed the order of Ce/MCM-48 (64%) > Ce/MCM-41 (54%) > MCM-48 (24%) > single ozonation (23%). They found that the active sites were surface protonated hydroxyl groups from the influence of initial pH and Ce/MCM-48 catalysts generated more hydroxyl radicals than Ce/MCM-41. Moreover, they investigated the degradation pathway of CA at different pH [131] (Figure 12). Roshani et al. [132] studied the catalytic ozonation of benzotriazole (BTZ) using $Mn/Al_2O_3$, $Cu/Al_2O_3$ and $Mn-Cu/Al_2O_3$ as catalyst, respectively, they found that a higher level of mineralization of BTZ rather than in uncatalyzed ozonation over a wide range of pH values. In deionized water, the catalytic activity followed the order of $Cu/Al_2O_3$ > $Mn-Cu/Al_2O_3$ > $Mn/Al_2O_3$ > Ozone. At pH 2, the order of catalytic activities was as follows: $Mn/Al_2O_3$ > $Mn-Cu/Al_2O_3$ > $Cu/Al_2O_3$ > Ozone. At pH 10, the catalytic activities obeyed the order of $Mn-Cu/Al_2O_3$ > $Mn/Al_2O_3$ > Ozone > $Cu/Al_2O_3$.

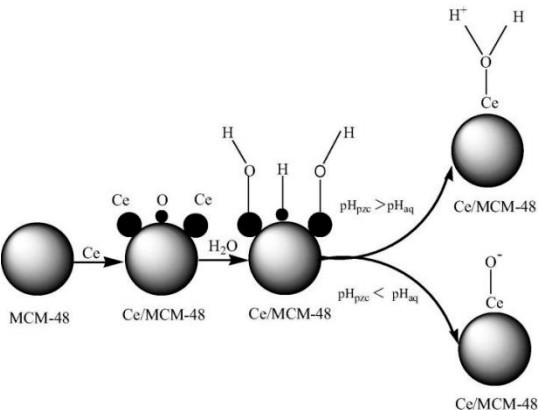

**Figure 12.** The probable reaction route of Ce/MCM-48 among different pH.

**Table 9.** Metals or metal oxides supported on oxides for catalytic ozonation studies.

| Catalyst | Pollutants | Operation Conditions | | Comments | References |
|---|---|---|---|---|---|
| $MnO_x$/sewage sludge-derived activated carbon ($MnO_x$/SAC) | Oxalic acid | [Cat] = 100 mg/L [Pull]$_0$ = 80 mg/L [Time] = 60 min | [$O_3$] = 5.0 mg/L pH = 3.5 [%] = 92.2% | The reaction mechanism involved both surface reactions and reactions in the bulk water, but dominantly surface reactions. | Huang et al. (2017) [130] |
| Ce/MCM-48 and Ce/MCM-41 | Clofibric acid | [Cat] = 0.4 mg/L [Pull]$_0$ = 10 mg/L [Time] = 120 min Ce/MCM-48: [%] = 64% Ce/MCM-41: [%] =54% | [$O_3$] = 1.7 mg/min pH = 4 | Addition of phosphate and sodium hydrogen sulfite into the reaction indicated hydroxyl radical mechanism for catalytic ozonation. | Li et al. (2017) [131] |
| $Fe_2O_3$/$Al_2O_3$@SBA-15 | Ibuprofen | [Cat] = 1.25 g/L [Pull]$_0$ = 10 mg/L [Time] = 60 min | [$O_3$] = 30 mg/L pH = 7.0 [%] = 90% | The $Fe_2O_3$/$Al_2O_3$@SBA-15 had high catalytic activity for Ibuprofen due to the surface oxygen atom, $\cdot OH_{ads}$ and $O_2^{\bullet-}$. | Bing et al. (2015) [133] |
| $Pr_6O_{11}$/$SiO_2$ @$Fe_3O_4$ | Acetochlor | [Cat] = 0.5 g/L [Pull]$_0$ = 20 mg/L [Time] = 120 min | [$O_3$] = 60 mL/min [%] = 37.3% | $Pr_6O_{11}$/$SiO_2$ @$Fe_3O_4$ was proved to be stable and recyclable. The catalytic ozonation process followed an $\cdot OH$ reaction mechanism. | Wang et al. (2018) [134] |
| Ag/$MnFe_2O_4$ | Di-n-butyl phthalate (DBP) | [Cat] = 10 mg/L [Pull]$_0$ = 0.5 mg/L [Time] = 60 min | [$O_3$] = 0.68 mg/min pH = 7.3 [%] = 75.3% | Ag/$MnFe_2O_4$ had highly porous structure with good magnetic property. Catalytic ozonation accelerated ozonation of DBP compared to the ozone-alone and undoped $MnFe_2O_4$ systems due to the increase of density of surface hydroxyl groups and electron transfer and cycle. | Wang et al. (2018) [135] |
| mFe/Cu | P-Nitrophenol (PNP) | [Cat] = 20 g/L [Pull]$_0$ = 500 mg/L [Time] = 30 min | [$O_3$] = 5.42 mg/L pH = 5.4 [%] = 93.6% | The reaction mechanism of the mFe/Cu/$O_3$ included catalytic ozonation, Fenton-like and/or peroxone reaction, adsorption and coagulation. | Xiong et al. (2018) [136] |
| Ni/$Al_2O_3$ | Succinic acid (SA) | [Cat] = 10 g/L [Pull]$_0$ = 200 mg/L [Time] = 60 min | [$O_3$] = 300 mL/min pH = 8.0 [%] = 100% | The preparation parameters and operational parameters had an effect on catalytic ozonation. Catalytic ozonation occurred via hydroxyl radical mechanism. | Peng et al. (2018) [137] |
| $Fe_2O_3$/AC | OA | [Cat] = 0.71 g/L [Pull]$_0$ = 30 mg/L [Time] = 60 min | [$O_3$] = 0.8 mg/min pH = 3.3 [%] = 89.2% | Acidic condition benefited OA removal in the $Fe_2O_3$/AC/$O_3$ process. A hydroxyl radical mechanism was proven in catalytic ozonation. | Li et al. (2018) [138] |
| $MnO_x$/SBA-15 | Norfloxacin | [Cat] = 0.1 g/L [Pull]$_0$ = 10 mg/L [Time] = 60 min | [$O_3$] = 1.7 mg/min pH = 5.0 [%] = 54% | Toxicological tests showed that a high detoxification was achieved after 30 min. | Chen et al. (2017) [139] |
| MgO/ceramic honeycomb (MgO/CH) | Acetic acid | [Cat] = 20 g/L [Pull]$_0$ = 100 mg/L [Time] = 30 min | [$O_3$] = 45.5 mg/min [%] = 81.6% | MgO/CH had a good reusability property by recycling test. Catalytic ozonation was governed by hydroxyl radical mechanism. | Shen et al. (2017) [140] |
| MWCNTs/$Fe_3O_4$ | Bisphenol A (BPA) | [Cat] = 0.5 g/L [Pull]$_0$ = 50 mg/L [Time] = 40 min | [$O_3$] = 3.0 mg/L pH = 7.0 [%] = 90% | MWCNTs/$Fe_3O_4$ had excellent catalytic activity, simple separation and good stability. | Huang et al. (2017) [141] |
| $Fe_3O_4$/multi-wall carbon nanotubes | Dimethyl phthalate (DMP) | [Cat] = 0.3 mg/L [Pull]$_0$ = 20 mg/L [Time] = 30 min | [$O_3$] = 4.8 mg/min pH = 6.8 [%] = 96% | The acidic sites of catalyst benefited ozone decomposition. $Fe_3O_4$ crystal structure was stable after five runs. | Bai et al. (2016) [142] |
| Mn-Fe/$Al_2O_3$ | BPA | [Cat] = 5.0 g/L [Pull]$_0$ = 50 mg/L [Time] = 30 min | [$O_3$] = 3.2 mg/min pH = 7 [%] = 84.1% | Hydroxyl radicals played a vital role in catalytic ozonation. Mn-Fe/$Al_2O_3$ exhibited good reusability and stability. | Liu et al. (2016) [143] |

**Table 9.** *Cont.*

| Catalyst | Pollutants | Operation Conditions | | Comments | References |
|---|---|---|---|---|---|
| Fe-Ni/AC | 2,4-Dichlorophenoxyacetic acid (2,4-D) | [Cat] = 0.5 g/L [Pull]$_0$ = 10 mg/L [Time] = 60 min | [O$_3$] = 0.8 mg/min pH = 4.2 [%] = 72% | The degradation rate constant of 2,4-D with Fe-Ni/AC/O$_3$ was 1.6 times higher than that with AC/O$_3$ and 1.9 times than that with ozonation alone. Degradation of 2,4-D followed the pseudo first order reaction model. | Lu et al. (2015) [144] |
| MnO$_2$-CuO/γ-Al$_2$O$_3$ | Ibuprofen and Humic Acid | [Cat] = 1.0 g/L Ibuprofen: [Pull]$_0$ = 5 mg/L Humic Acid: [Pull]$_0$ = 15 mg/L [Time] = 60 min Ibuprofen: [%] = 55% Humic Acid: [%] = 75% | [O$_3$] = 6.4 g/min pH = 5.6 | The noncatalytic mineralization increased by 10% due to the presence of humic acid. Adsorption played a major role in catalytic ozonation process. | Bibiana et al. (2015) [145] |
| Cu-Mn/γ-Al$_2$O$_3$ | Acid Red B | [Cat] = 4.0 g/L [Pull]$_0$ = 250 mg/L [Time] = 20 min | [O$_3$] = 4.26 mg/min pH = 8.5 [%] = 99.35% | Cu-Mn/γ-Al$_2$O$_3$ catalytic ozonation of Acid Red B followed the pseudo-first-order kinetics reaction model. The reaction followed hydroxyl radical mechanism. | Li et al. (2014) [146] |
| MnO$_x$/γ-Al$_2$O$_3$/TiO$_2$(MAT) | 4-chlorophenol (4-CP) | [Cat] = 2.0 g/L [Pull]$_0$ = 100 mg/L [Time] = 100 min | [O$_3$] = 2.0 mg/L pH = 6.6 [%] = 94.1% | 4-CP was oxidized primarily by hydroxyl radical mechanism. | Qi et al. (2014) [147] |
| Ni/TiO$_2$ | 2,4-D | [Cat] = 0.1 g/L [Pull]$_0$ = 80 mg/L [Time] = 20 min | [O$_3$] = 25 mg/L pH = 3.1 [%] = 97% | Ni/TiO$_2$ had high catalytic activity in catalytic ozonation for the mineralization of 2,4-D due to the synergic effect between ·OH and O$_3$. | Rodríguez et al. (2013) [148] |

## 6. Carbon Materials

To our best knowledge, activated carbon combined with ozone can provide better a removal efficiency of organic pollutes (Table 10). Active carbon behaves as the adsorbent and catalyst in promoting ozone oxidation. As said before, several authors have reported that activated carbon enhanced the decomposition of ozone [149–153]. The interest in the use of this material in ozonation processes has been increasing recently due to the good results observed [154–157]. The mechanism of catalytic ozonation behaves as a combination of bulk and surface reactions due to the presence of carbon materials by the addition of the radical scavenger tert-butanol [158,159].

In water, ozone decomposes into hydroxyl radicals (Equation (50)). The reactions between ozone, organic matter, catalyst, and aqueous solution in the presence of carbon materials (CMs) are as follows (Equations (51)–(57)) [160]:

$$O_3 \xrightarrow{OH^-} \cdot OH \tag{50}$$

$$CMs + Organic \rightarrow CMs - Organic \tag{51}$$

$$O_3 \xrightarrow{CMs} \cdot OH \tag{52}$$

$$\cdot OH + Organic \rightarrow Product \tag{53}$$

$$O_3 + CMs \rightarrow CMs - O \tag{54}$$

$$CMs - O + CMs - Organic \rightarrow Product \tag{55}$$

$$CMs - Organic + O_3 \rightarrow Product \tag{56}$$

$$CMs - Organic + \cdot OH \rightarrow Product \tag{57}$$

Gu et al. [161] investigated the oxidation of p-nitrophenol (PNP) by ozonation integrated with granular activated carbon (GAC). They found that the removal of organics was considerably enhanced due to the joint effect of oxidation and adsorption. Moreover, the results indicated that the mechanism was different at various pH. When at acidic conditions, the adsorption effect predominated in removing organics. On the contrary, when at basic conditions, the catalytic oxidation contributed primarily in organics removal. A sketch map of the main reaction on GAC at different pH conditions is shown in Figure 13 [161].

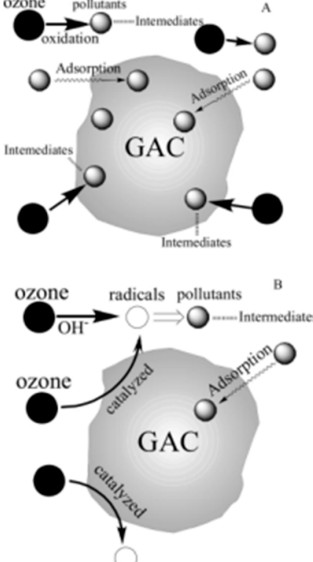

**Figure 13.** Sketch map of the main reaction on granular activated carbon (GAC) surface: (**A**) acidic conditions; (**B**) basic conditions.

Faria et al. [158] reported activated carbon (AC) for the degradation of oxamic and oxalic acids and found that a synergistic effect was observed in combing ozonation and AC for the oxidation of oxalic acid. To our best knowledge, AC enhanced the ozonation of both carboxylic acids due to its higher basicity, leading to a higher mineralization. At pH 3, the oxidation occurred mainly through reactions on the surface of the AC. While at pH 7, oxidation was included on surface reactions and bulk reactions. Moreover, with the increase of solution pH, the ozonation effectiveness of oxamic and oxalic acids decreased. At least in part, this was because of a higher solubility of solutes at neutral and basic and a consequent lower affinity of the AC surface. In other literature [22], the authors studied the catalytic ozonation of sulfonated aromatic compounds using AC as catalyst and found that the addition of AC enhanced the removal of sulfonated aromatic compounds due to the direct ozone reactions, adsorption and free radicals mechanisms.

Leili et al. [162] studied the catalytic ozonation of furfural in wastewater using granular activated carbon as catalyst, they found that 80.2% of furfural was removed via hydroxyl radical, while only 42.4% of furfural was removed by single ozonation. Moreover, the results shown that the degradation reaction of furfural occurred on the surface of the catalyst rather than in the solution. Guzman–Perez et al. [163] investigated the oxidation of atrazine by ozone mixed AC and found the reaction mainly occurred in the bulk liquid phase due to the inhibition effect of tert-butyl alcohol.

Wang et al. [164] prepared reduced graphene oxide (rGO) and used it for the catalytic ozonation of organic pollutants due to the higher defective level of rGO. Ozone molecules could decompose into active oxygen species on the structural vacancies and edges of rGO. Moreover, they found that the phenolic pollutants were vulnerable to direct ozone attacking, superoxide radicals and singlet oxygen and aliphatic organic pollutants were vulnerable to hydroxyl radicals.

**Table 10.** Carbon materials for catalytic ozonation studies.

| Catalyst | Pollutants | Operation Conditions | | Comments | References |
|---|---|---|---|---|---|
| MWCNTs | Sulfamethoxazole (SMX) | [Cat] = 0.14 g/L [Pull]$_0$ = 50 mg/L [Time] = 30 min | [O$_3$] = 50 mg/L pH = 4.8 [%] = 100% | The MWCNTs with various surface chemical properties were synthesized by oxidative and thermal treatments. MWCNTs significantly promoted the mineralization degree compared to ozonation alone. | Gonçalves et al. (2013) [155] |
| Reduced graphene oxide (rGO) | P-Hydroxylbenzoic Acid (PHBA) | [Cat] = 0.2 g/L [Pull]$_0$ = 5 mg/L [Time] = 60 min | [O$_3$] = 20 mg/L pH = 3.5 [%] = 95% | The reactive oxygen species (ROS) including superoxide radical ($\cdot$O$_2$$^-$) and singlet oxygen ($^1$O$_2$) were responsible for PHBA degradation in catalytic ozonation process. The electron-rich carbonyl groups were acted as the active sites for the catalytic reaction. | Wang et al. (2016) [165] |
| Carbon nanotubes (CNTs) | Methyl orange (MO) | [Cat] = 10 mg/L [Pull]$_0$ = 20 mg/L [Time] = 2 min | [O$_3$] = 2 mg/L pH = 3.0 [%] = 61% | The degradation of MO increased with pH from 2 to 3, while a reverse trend with the pH increased from 3 to 9. MO oxidation in solution occurred via molecular ozone. | Tizaoui et al. (2015) [166] |
| Multi-walled carbon nanotubes (MWCNT) | Oxalic acid | [Cat] = 0.14 mg/L [Pull]$_0$ = 90 mg/L | pH = 3.0 | The ball-milled MWCNT exhibited better results for the degradation of oxalic acid compared to the unmilled MWCNT. | Soares et al. (2015) [167] |
| MWCNT | Oxalic acid | [Cat] = 0.1 g/L [Pull]$_0$ = 90 mg/L [Time] = 40 min | [O$_3$] = 20 mg/min pH = 3.0 [%] = 79.4% | Catalyst dosage and the reaction temperature showed positive effects on the removing of oxalic acid in catalytic ozonation. With the increase of initial pH from 1.0–3.0, the oxalic acid removal increases, in contrast, decreasing with further increasing of pH from 3.0 to 6.1. | Liu et al. (2011) [168] |

## 7. Minerals Modified with Metals

Low cost natural solid minerals, such as sand, soils, zeolites, red mud, and goethite have been recently used in the heterogeneous catalytic ozonation of organic pollutants (Table 11).

Chen et al. [169] investigated the catalytic ozonation of nitrobenzene using ZSM5 zeolites loaded with metallic (Ce, Fe, or Mn) oxides as catalyst. The removal efficiency of total organic carbon (TOC) with NaZSM5-38, HZSM5-38, and NaZSM5-100 was 45.9%, 62.3%, 59.0%, while the removal (39.2%) by single ozonation. The Ce/NaZ38 zeolite had the highest catalytic activity in removing TOC (86.3%) among all ZSM5 zeolites studies. However, Ikhlaq et al. [170] found that the zeolites with low $SiO_2/Al_2O_3$ ratios owned a high adsorption capacity in removing ibuprofen. At acidic pH, the catalyst had the highest activity for ibuprofen, while it was ineffective for removing acetic acid. It was found that catalytic ozonation of organic pollutants using ZSM-5 zeolites occurred via the direct reactions of molecular ozone with pollutants [170] (Figure 14). In other work, Ikhlaq et al. [28] proved again the ability to adsorb ozone and coumarin of ZSM-5 zeolites. Zeolites acted as reservoirs of ozone and adsorbents of organic compounds [171]. More importantly, they found that the activity was independent of zeolite acidity and dependent on the hydrophobicity of the zeolite.

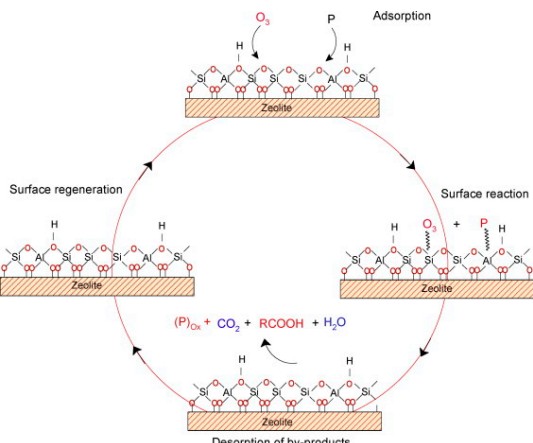

**Figure 14.** Proposed mechanism of catalytic ozonation on ZSM-5 zeolites (P, Pollutants).

Valdés et al. [172] reported that the acid treatment improved the catalytic ozonation activity of natural zeolite for methylene blue (MB) due to the decrease the $pH_{PZC}$ and increase the Brønsted acidity. It was found that Brønsted acid sites played an important role in catalytic ozonation.

Yuan et al. [173] investigated the catalytic ozonation of p-chloronitrobenzene (p-CNB) using Fe/pumice as catalyst, they found that the addition of Fe/pumice increased the removal efficiency of p-CNB. In the ozonation reaction, the main oxide species were hydroxyl radicals. The uncharged surfaces hydroxyl groups were beneficial to the catalytic activity of the Fe/pumice [28]. Gao et al. [174] reported that 73.3% of diclofenac (DCF) was removed in iron silicate-loaded pumice (FSO/PMC)/$O_3$ process due to the FSO/PMC which enhanced the mass transfer and solubility of aqueous ozone, and accelerated the production of · OH radicals.

Xu et al. [175] studied the cobalt-loaded red mud (RM) catalytic ozonation of bezafibrate (BZF) degradation. The contributions of 39.22% of molecular ozone, 45.20% of hydroxyl radicals, and 15.57% of surface adsorption to BZF degradation were identified, confirming that radical oxidation played an important role in the catalytic ozonation. Li et al. [176] studied the efficiency, intermediates, and toxicity in this process. Xu et al. [177] reported cerium-modified red mud (RM) catalysts in the catalytic ozonation of BZF. Ce (IV)/RM-p showed the best performance for removing BZF due to the hydroxyl radical, and superoxide ions were a vital key for ozone decomposition in catalytic ozonation using Ce (IV)/RM-p.

**Table 11.** Minerals modified with metals for catalytic ozonation studies.

| Catalyst | Pollutants | Operation Conditions | | Comments | References |
|---|---|---|---|---|---|
| Iron silicate-loaded pumice (FSO/PMC) | Diclofenac (DCF) | [Cat] = 0.8 g/L [Pull]$_0$ = 29.6 mg/L [Time] = 60 min | [O$_3$] = 5.52 mg/L pH = 7.0 [%] = 73.3% | The DCF mineralization was enhanced in FSO/PMC catalytic ozonation process due to the improvement of mass transfer of aqueous ozone, increase of the solubility of aqueous ozone, and acceleration of the generation of ·OH radicals. | Gao et al. (2017) [173] |
| Cobalt-loaded red mud (RM) | Bezafibrate (BZF) | [Cat] = 50 mg/L [Pull]$_0$ = 10 mg/L [Time] = 30 min | [O$_3$] = 0.5 mg/L pH = 6.5 | Surface cobalt loading contributed to the change of the structure, surface chemical properties and catalytic activity of Co/RM. Degradation of BZF followed hydroxyl radical mechanism. | Xu et al. (2016) [175] |
| LaCoO$_3$ | Benzotriazole (BZA) | [Cat] = 0.5 mg/L [Pull]$_0$ = 10 mg/L [Time] = 15 min | [O$_3$] = 2.0 mg/L pH = 6.4 [%] = 100% | The surface hydroxyl groups of LaCoO$_3$ accelerated the decomposition of ozone to generate more radicals. | Zhang et al. (2018) [178] |
| Tourmaline | Atrazine (ATZ) | [Cat] = 1.0 g/L [Pull]$_0$ = 1.1 mg/L [Time] = 10 min | [O$_3$] = 3.0 mg/L pH = 7.0 [%] = 98% | Catalytic ozonation using tourmaline resulted in higher ATZ removal efficiency compared to single ozonation. | Wang et al. (2018) [179] |
| Natural mackinawite (NM) | *N,N*-dimethylacetamide (DMAC) | [Cat] = 3.5 g/L [Time] = 20 min [%] = 95.4% | [O$_3$] = 0.3 L/min pH = 6.8 | Degradation of DMAC was governed by hydroxyl radical mechanism. | Peng et al. (2018) [180] |
| ZSM-5 Zeolites | Nitrobenzene | [Cat] = 1.0 g/L [Pull]$_0$ = 100 mg/L [Time] = 50 min | [O$_3$] = 5 mg/min pH = 7.2 [%] = 74% | NaZSM-5 catalytic ozonation of nitrobenzene by adsorption and direct ozonation for the first use and direct ozonation and ●OH mediated oxidation for after eight recycles. The more Si-O bonds on zeolite surfaces contributed to the higher catalytic activity of NaZSM-5. | Wang et al. (2018) [181] |
| Zeolite4A (Z4A) | Paracetamol | [Cat] = 11 g/L [Pull]$_0$ = 50 mg/L [Time] = 60 min | [O$_3$] = 0.9 mg/min pH = 7.12 [%] = 90.68% | The Z4A did not promote the decomposition of ozone to produce superoxide ion radical and hydroxyl radicals. It was found that the catalytic ozonation reaction followed a non-radical mechanism. | Ikhlaq et al. (2018) [182] |
| Clinoptilolite | Nalidixic acid (NA) | [Cat] = 6 g/L [Pull]$_0$ = 20 mg/L [Time] = 60 min | [O$_3$] = 0.25 mg/min pH = 7.0 [%] = 73.8% | The hydroxyl and superoxide radicals, adsorption and catalyst active sites played a vital role in catalytic ozonation of NA. | Khataee et al. (2017) [183] |
| Fe/pumice | P-chloronitrobenzene (p-CNB) | [Cat] = 0.5 g/L [Pull]$_0$ = 0.1 mg/L [Time] = 15 min | [O$_3$] = 0.9 mg/L pH = 6.0 [%] = 90.8% | The uncharged surfaces hydroxyl groups were responsible for catalytic activity of the Fe/pumice. Degradation of p-CNB followed hydroxyl radical mechanism | Yuan et al. (2016) [184] |
| Red mud (RM) | Nitrobenzene | [Cat] = 0.5 g/L [Pull]$_0$ = 1.0 mg/L [Time] = 40 min | [O$_3$] = 1.0 mg/L pH = 7.0 | Hydroxyl radical played a role in nitrobenzene degradation. | Qi et al. (2014) [185] |
| Raw bauxite | 2,4,6-trichloroanisole (TCA) | [Cat] = 0.2 g/L [Pull]$_0$ = 1 × 10$^{-4}$ mg/L [Time] = 10 min | [O$_3$] = 0.5 mg/L pH = 6.0 [%] = 95.2% | Presence of the raw bauxite in ozonation improved the degradation of TCA. Producing ●OH in the catalytic ozonation process due to the introduction of the raw bauxite. | Qi et al. (2009) [186] |
| Y zeolite | Phenol | [Cat] = 4.2 g/L [Pull]$_0$ = 100 mg/L [Time] = 45 min | [O$_3$] = 0.3 mg/min [%] = 50.9% | Y zeolite enhanced the decomposition of ozone and the production of hydroxyl. Y zeolite showed excellent repetitive-use performance even after 10 runs of experiments. | Dong et al. (2008) [187] |

## 8. Conclusions and Future Perspective

The review presented the application of catalysts in ozonation processes, and the removal efficiency of pollutants in aqueous media was summarized. Now, many materials modified with metals, metal oxides, bimetals, minerals, activated carbon, different precursors or a combination of these should be tried out in order to analyze their effect on the degradation or mineralization of contaminant. The quality of water, kind of organic pollutant, type of catalyst, its surface performance, performance of the surface-active sites, the pH of the solution effect, and the reaction of ozone decomposition in solutions play an important role in removing contaminants from wastewater. It is worth mentioning that every group of catalysts should be investigated, analyzed, and discussed separately in catalytic ozonation due to the variety of surface performances of the catalysts and interactions of the catalysts, ozone, and organic pollutants. However, the reaction mechanism of catalytic ozonation is not clear. Its applications were mainly limited to the laboratory field. In many papers, researchers have failed to control pH after adding the catalyst into the water and neglected the competing adsorption of water molecules on the ozone decomposition sites.

In order to deeply and judiciously analyze the application of catalytic ozonation process and provide theoretical support for its commercial application, the use of catalysts in ozonation processes could rely on practical situations. Environmental friendliness, better catalytic activity and good stability, and heterogeneous catalysts are an absolute need. Deeper knowledge of the application of catalysts in ozonation processes is important to design a suitable catalyst to promote the mineralization of the organic contaminants.

**Author Contributions:** Project administration, B.W.; writing—original draft preparation, H.Z.; writing—review and editing, F.W. and X.X.; comments and suggestions, B.W.; data curation—K.T., Y.S. and T.Y.

**Funding:** This research was funded by Open Fund of State Key Laboratory of Oil and Gas Reservoir Geology and Exploitation (Southwest Petroleum University), grant number PLN1125 and the National Science and Technology Major Project of China, grant number 2016ZX05062.

**Acknowledgments:** This work was carried out with the financial supports of Open Fund of State Key Laboratory of Oil and Gas Reservoir Geology and Exploitation (Southwest Petroleum University) (Grant No. PLN1125) and the National Science and Technology Major Project of China (Grant No. 2016ZX05062).

**Conflicts of Interest:** The authors declare no conflict of interest.

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
