# Peer review of "Application of Heterogeneous Catalytic Ozonation for Refractory Organics in Wastewater"

_catalysts, doi:10.3390/catal9030241_

Round 1
Reviewer 1 Report
The topic is interesting enough for the journal, and the material collected for this mamnuscript is appropriate for a review article. However, its style, composition and English needs considerable improvements.
On the basis of the whole text of the manuscript, the title seems to be not quite suitable. It emphasizes "mechanism", but the work itself discusses various types of catalysts applied in combination with ozonation, but not necesserily from thom the viewpoint mechanism.
Its English needs many corrections regarding the errors of grammar (typical error is, e.g., singular vs. plural problem) and, in some cases, also. in the respect of style. Several of them are highlighted in the attached manuscript. It also contains some remarks and questions regarding the scientific content, too.
As to the organization of the manuscript, it is not quite consistent. In some cases, certain works in the literature are discussed in the text and listed in a corresponding table as well, repeated the related pieaces of information, while other works are just shown in a table, although they may be more important than the previously mentioned ones. What was the concept in this respect?
The quality (resolution) of several figures is rather poor - probably due to a not suitable procedure of taking them from the original articles.

Reviewer 2 Report
The mechanism of heterogeneous catalytic ozonation of refractory organics in wastewater
This paper must be improved in several aspects, but I believe that the authors are able to do really god job and significantly append the review with necessary data and discussion.
1. Title should be changed – this paper is not focusing enough on “The mechanism of ….” – this aspect is only mentioned.
2. Other AOPs using ozone in combined processes should be mentioned in the introduction, this papers will be useful:
a. M. Gagol et al. Wastewater treatment by means of advanced oxidation processes based on cavitation - A Review, Chem. Eng. J. 2018
b. G. Boczkaj, Wastewater treatment by means of Advanced Oxidation Processes at basic pH conditions: A review, Chem. Eng. J. 2017
3. Data in the tables should be presented and compared in same aspects. For example table 4. -> each paper is discussed using a “highlights of the work” – the authors based on highlights (bullet points) provided for most of the manuscript? For some the method of synthesis is provided, for other temperature of calcination for other rate constant or order of the reaction – what it gives for the readers? Nothing. Please compare “comparable” aspects like % effectiveness (for each paper); catalyst optimal conc.; What was the increase of degradation of non catal.-catal. Process? What was the molar ratio of ozone to pollutant (if the original provide it in different manner – please make necessary calculations). Provide pH of the treatment. I think more columns in the tables is needed.
4. The authors should calculate a rox (molar ratio of oxidant to pollutant) and discuss it in terms of obtained %degradation.
5. Lines 569-574 and above. It should be explained shortly how authors of different papers confirmed the main place of the reaction i.e. bulk vs surface. It will be helpful for readers.
6. I understand that the authors focused on heterogenic catalysis, but for future researchers (especially the young scientists) it should be clearly stated that there are other methods of enhancing ozone effectiveness, including
· Photocatalytic processes
· Peroxone technology –this papers will be useful:
a. 2017, Study of Different Advanced Oxidation Processes for Wastewater Treatment from Petroleum Bitumen Production at Basic pH, Ind. Eng. Chem. Res. 56, 8806-8814
b. 2019, Pilot scale degradation study of 16 selected volatile organic compounds by hydroxyl and sulfate radical based advanced oxidation processes, J. Clean. Prod.
· Cavitation based processes:
a. 2018, Highly effective degradation of selected groups of organic compounds by cavitation based AOPs under basic pH conditions, Ultrason. Sonochem. 45, 257-266.
b. 2018, Effective method of treatment of industrial effluents under basic pH conditions using acoustic cavitation – a comprehensive comparison with hydrodynamic cavitation processes, Chem. Eng. Process. 128, 103-113
c. 2018, Effective method of treatment of effluents from production of bitumens under basic pH conditions using hydrodynamic cavitation aided by external oxidants, Ultrason. Sonochem. 40, 969-979
These alternatives should be highlighted along with a short discussion of advantages and disadvantages in comparison to heterogeneous catalytic ozonation.
7. Catalyst re-usability is not compared – it should.
8. The effect of inorganic ions should be discussed in details.
9. Conclusion part must be re-writed – some aspects are listed twice.
10. Table 2: “Lodine”. “TiO2+hv” – it relates to elektron excitation or to h+ (?) This should be explained under the table as well as in par. 4.2
11. Line 858: “80. 80. Zhang, T.;”
12. The authors should highlight the risk of formation and importance of monitoring of oxygenated volatile organic compounds in AOPs, including ozonation. This relates also to biotoxicity of the final effluent. These papers will be useful to support this statement:
a. 2014, New Procedures for Control of Industrial Effluents Treatment Processes, Ind. Eng. Chem. Res. 53 (4), 1503–1514
b. 2016, Application of dynamic headspace and gas Chromatography coupled to mass spectrometry (DHS-GC-MS) for the determination of oxygenated volatile organic compounds in refinery effluents, Anal. Methods, 8, 3570-3577.
c. 2016, Application of dispersive liquid-liquid microextraction and gas chromatography with mass spectrometry for the determination of oxygenated volatile organic compounds in effluents from the production of petroleum bitumen, J. Sep. Sci. 39, 2604-15
d. P. Makoś, et al., Method for the determination of carboxylic acids in industrial effluents using dispersive liquid-liquid microextraction with injection port derivatization gas chromatography–mass spectrometry, J. Chromatogr. A 2017, 1517, 26-34
e. P. Makoś, et al., Sample preparation procedure using extraction and derivatization of carboxylic acids from aqueous samples by means of deep eutectic solvents for gas chromatographic-mass spectrometric analysis, J. Chromatogr. A 2018, 1555, 10-19
13. About biotoxicity – how many of reviewed papers relate to changes of biotoxicity during studied processes? What are the general conclusions?
14. Relate to the issue of metals leaching from the catalyst and their presence in final effluent.
15. The review should have some degree of criticism
16. The conclusions should provide some wisdom for other researchers
Round 2
Reviewer 1 Report
Most of the errors of grammar indicated earlier have been corrected. However, still, in several cases, there remained problems in this respect. Moreover, even the newly inserted texts contains some disturbing errors. The whole manuscript ought to be checked in this respect by a professional English lector.
Regarding the answers to the questions; they ought to be merged (briefly) into the manuscript, to make it more precise and understandable for the readers
Author Response
Reviewer #1: Most of the errors of grammar indicated earlier have been corrected. However, still, in several cases, there remained problems in this respect. Moreover, even the newly inserted texts contains some disturbing errors. The whole manuscript ought to be checked in this respect by a professional English lector.
Regarding the answers to the questions; they ought to be merged (briefly) into the manuscript, to make it more precise and understandable for the readers
Response: We apologize for our errors presented in the original paper. We have carefully revised this paper by examining all wordings through several times, trying our best to reduce errors as many as possible.
Here below are all instances:
Before revised | After revised | |
Line 15 | have | has |
Line 28 | were | are |
Line 32 | were | are |
Line 48 | Hydrodynamic | hydrodynamic |
Line 52 | from | in |
matter | matters | |
Line 56 | emphasized | emphasizes |
Line 65 | amongst | among |
Line 66 | technology | technologies |
is | are | |
Line 71 | technology | technologies |
Line 72 | application | applications |
Line 76 | has | have |
Line 83 | own | own |
Line 106 | cataysts | catalysts |
Line 114 | were | are |
Line 115 | suggested | suggest |
was | is | |
Line 125 | It is | They are |
catalyst | catalysts | |
Line 130 | structure | structures |
Line 133 | has | had |
Line 136 | can | could |
Line 149 | is | was |
Line 151 | pHzpc | pHzpc |
Line 152 | has | had |
Line 153 | is | was |
Line 168 | are | were |
Line 215 | has | had |
Line 219 | relays | relayed |
Line 234 | exist | existed |
Line 236 | catalyst | catalysts |
Line 239 | is | was |
Line 244 | can | could |
Line 247 | is | was |
shows | shown | |
Line 265 | weakens | weakened |
Line 266 | weakens | weakened |
Line 270 | can | could |
Line 272 | will | were |
group | groups | |
is | were | |
Line 273 | illustrate | illustrated |
Line 275 | density | densities |
Line 278 | follow | followed |
Line 279 | follow | followed |
Line 291 | can | could |
was | were | |
Line 303 | catalyst | catalysts |
Line 304 | its | their |
activity | activities | |
Line 307 | enhances | enhanced |
Line 308 | decrease | decreased |
declines | declined | |
Line 314 | is | was |
effectivity | effective | |
Line 319 | is | was |
Line 322 | can | could |
Line 328 | was | were |
activity | activities | |
Line 339 | showed | shown |
performance | performances | |
Line 348 | were | was |
Line 376 | capability | capabilities |
Line 377 | dosage | dosages |
Line 378 | capability | capabilities |
was | were | |
Line 401 | activity | activities |
Line 405 | is | was |
Line 444 | promote | promoted |
Line 445 | enhance | enhanced |
Line 446 | performance | performances |
Line 449 | activity | activities |
have | had | |
Line 453 | includes | included |
Line 458 | consist | consisted |
Line 478 | was | were |
Line 483 | follows | followed |
Line 485 | is | was |
Line 498 | capability | capabilities |
Line 506 | activity | activities |
Line 508 | activity | activities |
Line 520 | behaved | behaves |
Line 539 | is | was |
Line 544 | is | was |
Line 547 | including | included |
Line 558 | atrazinem | atrazine |
Line 575 | studied | studies |
Line 576 | own | owned |
Line 577 | has | had |
They are | It was | |
Line 582 | is | was |
Line 592 | is | was |
Line 596 | accelerate | accelerated |
Line 609 | was | is |
Line 611 | its | their |
Line 617 | performance | performances |
Line 618 | Its application is | Their applications are |
Line 624 | Environmently | Environmentally |
Line 623 | its | their |
application | applications | |
Table 3 | is | was |
have | had | |
high | higher | |
Table 4 | follow | followed |
is | was | |
Table 5 | acted | act |
Table 6 | group | groups |
was | were | |
Table 7 | was | were |
Table 9 | including | included |
plays | played | |
Table 11 | do | did |
Reviewer 2 Report
I accept the manuscript in its present form.
Author Response
Reviewer #2: I accept the manuscript in its present form.
Thank you for your approval!